# SLIP: Learning to Predict in Unknown Dynamical Systems with Long-Term Memory

**Paria Rashidinejad**   **Jiantao Jiao**   **Stuart Russell**
EECS Department, University of California, Berkeley
{paria.rashidinejad, jiantao, russell}@berkeley.edu

## Abstract

We present an efficient and practical (polynomial time) algorithm for online prediction in unknown and partially observed linear dynamical systems (LDS) under stochastic noise. When the system parameters are known, the optimal linear predictor is the Kalman filter. However, in unknown systems, the performance of existing predictive models is poor in important classes of LDS that are only marginally stable and exhibit long-term forecast memory. We tackle this problem by bounding the generalized Kolmogorov width of the Kalman filter coefficient set. This motivates the design of an algorithm, which we call spectral LDS improper predictor (SLIP), based on conducting a tight convex relaxation of the Kalman predictive model via spectral methods. We provide a finite-sample analysis, showing that our algorithm competes with the Kalman filter in hindsight with only logarithmic regret. Our regret analysis relies on Mendelson's small-ball method, providing sharp error bounds without concentration, boundedness, or exponential forgetting assumptions. Empirical evaluations demonstrate that SLIP outperforms state-of-the-art methods in LDS prediction. Our theoretical and experimental results shed light on the conditions required for efficient probably approximately correct (PAC) learning of the Kalman filter from partially observed data.

## 1   Introduction

Predictive models based on linear dynamical systems (LDS) have been successfully used in a wide range of applications with a history of more than half a century. Example applications in AI-related areas range from control systems and robotics [11] to natural language processing [4], healthcare [41], and computer vision [6, 7]. Other applications are found throughout the physical, biological, and social sciences in areas such as econometrics, ecology, and climate science.

The evolution of a discrete-time LDS is described by the following state-space model with $t \geq 1$:

$$h_{t+1} = Ah_t + Bx_t + \eta_t,$$
$$y_t = Ch_t + Dx_t + \zeta_t,$$

where $h_t$ are the latent states, $x_t$ are the inputs, $y_t$ are the observations, and $\eta_t$ and $\zeta_t$ are process and measurement noise, respectively.

When the system parameters are known, the optimal linear predictor is the Kalman filter. When they are unknown, a common approach for prediction is to first estimate the parameters of a Kalman filter and then use them to predict system evolution. Direct parameter estimation usually involves solving a non-convex optimization problem, such as in the expectation maximization (EM) algorithm, whose theoretical guarantees may be difficult [55]. Several recent works have studied finite-sample properties of LDS identification. For fully observed LDS, system identification is shown to be possible without a strict stability ($\rho(A) < 1$) assumption, where $\rho(A)$ is the spectral radius of $A$ [46, 43, 13].

For partially observed LDS, methods such as gradient descent [19] and subspace identification [49] are developed, whose performances degrade polynomially when $\rho(A)$ is close to one.

We focus on constructing LDS *predictors* without identifying the parameters. For a stochastic LDS, the recent work of Tsiamis and Pappas [48] is most related to our question. Their method performs linear regression over a fixed-length lookback window to predict the next observation $y_t$ given its causal history. Without using a mixing-time argument, [48] showed logarithmic regret with respect to the Kalman filter in hindsight even when the system is marginally stable ($\rho(A) \leq 1$). However, the prediction performance deteriorates if the true Kalman filter exhibits *long forecast memory*.

To illustrate the notion of forecast memory, we recall the recursive form of the (stationary) Kalman filter for $1 \leq t \leq T$, where $T$ is the final horizon [25, Chapter 9]:

$$\hat{h}_{t+1|t} = A\hat{h}_{t|t-1} + Bx_t + K(y_t - C\hat{h}_{t|t-1} - Dx_t) \tag{1}$$

$$= (A - KC)\hat{h}_{t|t-1} + Ky_t + (B - KD)x_t, \tag{2}$$

where $\hat{h}_{t|t-1}$ denotes the optimal linear predictor of $h_t$ given all the observations $y_1, y_2, \ldots, y_{t-1}$ and inputs $x_1, x_2, \ldots, x_{t-1}$. The matrix $K$ is called the (predictive) Kalman gain.[1] The Kalman predictor of $y_t$ given $y_1, y_2, \ldots, y_{t-1}$ and $x_1, x_2, \ldots, x_t$, denoted by $\hat{y}_{t|t-1}$, is $C\hat{h}_{t|t-1} + Dx_t$. Assume that $\hat{h}_{1|0} = 0$. By expanding Equation (2), we obtain

$$m_t \triangleq \hat{y}_{t|t-1} = \sum_{i=1}^{t-1} CG^{t-i-1}Ky_i + \sum_{i=1}^{t-1} CG^{t-i-1}(B - KD)x_i + Dx_t, \tag{3}$$

where $G = A - KC$. In an LDS, the transition matrix $A$ controls how fast the process mixes—i.e., how fast the marginal distribution of $y_t$ becomes independent of $y_1$. However, it is $G$ that controls how long the *forecast* memory is. Indeed, it was shown in [25, chap. 14] that if the spectral radius $\rho(G)$ is close to one, then the performance of a linear predictor that uses only $y_{t-k}$ to $y_{t-1}$ for fixed $k$ in predicting $y_t$ would be substantially worse than that of a predictor that uses all information $y_1$ up to $y_{t-1}$ as $t \to \infty$. Conceivably, the sample size required by the algorithm of Tsiamis and Pappas [48] explodes to infinity as $\rho(G) \to 1$, since the predictor uses a fixed-length lookback window to conduct linear regression.

The primary reason to focus on long-term forecast memory is the ubiquity of long-term dependence in real applications, where it is often the case that not all state variables change according to a similar timescale[2] [5]. For example, in a temporal model of the cardiovascular system, arterial elasticity changes on a timescale of years, while the contraction state of the heart muscles changes on a timescale of milliseconds; see Appendix I for a discussion on systems with long forecast memory.

Designing provably computationally and statistically efficient algorithms in the presence of long-term forecast memory is challenging, and in some cases, impossible. A related problem studied in the literature is the prediction of auto-regressive model with order infinity: $\text{AR}(\infty)$. Without imposing structural assumptions on the coefficients of an $\text{AR}(\infty)$ model, there is no hope to guarantee vanishing prediction error. One common approach to obtain a smaller representation is to make an exponential forgetting assumption to justify finite-memory truncation. This approach has been used in approximating $\text{AR}(\infty)$ with decaying coefficients [18], LDS identification [19], and designing predictive models for LDS [48, 26]. Inevitably, the performance of these methods degrade by either losing long-term dependence information or requiring very large sample complexity as $\rho(G)$ (and sometimes, $\rho(A)$) gets closer to one.

However, the Kalman predictor in (3) does seem to have a structure and in particular, the coefficients are geometric in $G$, which gives us hope to exploit it. Our main contributions are the following:

**1. Generalized Kolmogorov width and spectral methods:** We analyze the *generalized Kolmogorov width* (defined in Section 5.1) of the Kalman filter coefficient set. In Theorem 2, we show that when the matrix $G$ is diagonalizable with *real* eigenvalues, the Kalman filter coefficients can be approximated

by a linear combination of $\mathrm{polylog}\,(T)$ *fixed known* filters with $1/\,\mathrm{poly}\,(T)$ error. It then motivates the algorithm design of linear regression based on the *transformed* features, where we first transform the observations $y_{1:t}$ and inputs $x_{1:t}$ for $1 \le t \le T$ via these fixed filters. In some sense, we use the transformed features to achieve a good bias-variance trade-off: the small number of features guarantees small variance and the generalized Kolmogorov width bound guarantees small bias. We show that the fixed known filters can be computed efficiently via spectral methods. Hence, we choose spectral LDS improper predictor (SLIP) as the name for our algorithm.

**2. Difficulty of going beyond real eigenvalues:** We show in Theorem 2 that if the dimension of matrix $G$ in (3) is at least 2, then without assuming real eigenvalues one has to use at least $\Omega(T)$ filters to approximate an arbitrary Kalman filter. In other words, the Kalman filter coefficient set is very difficult to approximate via linear subspaces in general. This suggests some inherent difficulty of constructing provable algorithms for prediction in an arbitrary LDS.

**3. Logarithmic regret uniformly for $\rho(G) \le 1, \rho(A) \le 1$:** When $\rho(A)$ or $\rho(G)$ is equal to one the process does not mix and common assumptions regarding boundedness, concentration, or stationarity do not hold. Recently, Mendelson [38] showed that such assumptions are not required and learning is possible under a milder assumption referred to as the *small-ball* condition. In Theorem 1, we leverage this idea as well as results on self-normalizing martingales and show a logarithmic regret bound for our algorithm uniformly for $\rho(G) \le 1$ and $\rho(A) \le 1$.

**4. Experimental results:** We demonstrate in simulations that our algorithm performs better than the state-of-the-art in LDS prediction algorithms. In Section 7, we compare the performance of our algorithm to wave filtering [22] and truncated filtering [48].

## 2   Related work

Adaptive filtering algorithms are classical methods for predicting observations without the intermediate step of system identification [34, 15, 16, 51, 29, 35]. However, finite-sample performance and regret analysis with respect to optimal filters are typically not studied in the classical literature. From a machine learning perspective, finite-sample guarantees are critical for comparing the accuracy and sample efficiency of different algorithms.

In designing algorithms and analyses for learning from sequential data, it is common to use mixing-time arguments [54]. These arguments justify finite-memory truncation [19, 18] and support generalization bounds analogous to those in i.i.d. data [39, 28]. An obvious drawback of mixing-time arguments is that the error bounds degrade with increasing mixing time. Several recent works established that identification is possible for systems that do not mix [46, 13, 47]. For the problem of the linear quadratic regulator, several recent results provided finite-sample regret bounds for fully-observed systems [12, 40, 8, 2, 37, 45] and partially-observed stable systems [44, 31, 33].

For prediction without LDS identification, Hazan et al. [22, 23] have proposed algorithms for the case of bounded adversarial noise. Similar to our work, they use spectral methods for deriving features. However, the spectral method is applied on a different set and connections with $k$-width and difficulty of approximation for the non-diagonalizable case are not studied. Moreover, the regret bounds are computed with respect to a certain fixed family of filters and competing with the Kalman filter is left as an open problem. Indeed, the predictor for general LDS proposed by Hazan et al. [23] without the real eigenvalue assumption only uses a fixed lookback window. Furthermore, the feature norms are of order $\mathrm{poly}\,(T)$ in our formulation, which makes a naive application of online convex optimization theorems [21] fail to achieve a sublinear regret.

We focus on a more challenging problem of learning to predict in the presence of unbounded stochastic noise and long-term memory, where the observation norm grows over time. Most related to our work are the recent works [48, 17], where the performance of an algorithm based on a finite lookback window is shown to achieve logarithmic regret with respect to the Kalman filter. However, the performance of this algorithm degrades as the forecast memory increases. In fact, this algorithm can be viewed as a special case of our algorithm where the fixed filters are chosen to be standard basis vectors.

We investigate the possibility of a tight convex relaxation of the Kalman predictive model by analyzing a generalization of Kolmogorov width. Kolmogorov width is a notion from approximation theory that measures how well a set can be approximated by a low-dimensional linear subspace

[42]. Kolmogorov width has been used in a variety of problems such as minimax risk bounds for truncated series estimators [10, 24], minimax rates for matrix estimation [36], density estimation [20], hypothesis testing [52, 53], and compressed sensing [9]. In Section 5, we present a generalization of Kolmogorov width, which facilitates measuring the convex relaxation approximation error.

## 3 Preliminaries and problem formulation

**Notation.** We denote by $x_{1:t} \in \mathbb{R}^{nt}$, the vertical concatenation of $x_1, \ldots, x_t \in \mathbb{R}^n$. We use $x_t(i)$ to refer to the $i$-th element of the vector $x_t = [x_t(1), \ldots, x_t(n)]^\top$. We denote by $\|.\|_2$, the Euclidean norm of vectors and the operator 2-norm of matrices. The spectral radius of a square matrix $A$ is denoted by $\rho(A)$. The eigenpairs of an $n \times n$ matrix are $\{(\sigma_j, \phi_j)\}_{j=1}^n$ where $\sigma_1 \geq \cdots \geq \sigma_n$ and $\{\phi_j\}_{j=1}^k$ are called the top $k$ eigenvectors. We denote by $\phi_j(t : 1) = [\phi_j(t), \ldots, \phi_j(1)]$ the first $t$ elements of $\phi_j$ in a reverse order. The Kronecker product of matrices $A$ and $B$ is denoted by $A \otimes B$. Identity matrix of dimension $n$ is represented by $I_n$. We write $x \lesssim_b y$ to represent $x \leq cy$, where $c$ is a constant that only depends on $b$. We use the notation $x \asymp_b y$ if $c_1, c_2 > 0$ exist that only depend on $b$ and $c_1|x| \leq |y| \leq c_2|x|$.

**Problem statement.** We consider the problem of predicting observations generated by the following linear dynamical system with inputs $x_t \in \mathbb{R}^n$, observations $y_t \in \mathbb{R}^m$, and latent states $h_t \in \mathbb{R}^d$:

$$
\begin{aligned}
h_{t+1} &= Ah_t + Bx_t + \eta_t, \\
y_t &= Ch_t + Dx_t + \zeta_t,
\end{aligned}
\tag{4}
$$

where $A, B, C$, and $D$ are matrices of appropriate dimensions. The sequences $\eta_t \in \mathbb{R}^d$ (process noise) and $\zeta_t \in \mathbb{R}^m$ (measurement noise) are assumed to be zero-mean, i.i.d. random vectors with covariance matrices $Q$ and $R$, respectively. For presentation simplicity, we assume that $\eta_t$ and $\zeta_t$ are Gaussian; extension of our regret analysis to sub-Gaussian and hypercontractive noise is straightforward. We assume that the discrete Riccati equation of the Kalman filter for the state covariance has a solution $P$ and the initial state starts at this stationary covariance. This assumption ensures the existence of the stationary Kalman filter with stationary gain $K$; see [25] for details.

Define the observation matrix $\mathcal{O}_t$ and the control matrix $\mathcal{C}_t$ of a stationary Kalman filter as

$$
\begin{aligned}
\mathcal{O}_t &= \begin{bmatrix} CG^{t-1}K & CG^{t-3}K & \ldots & CK \end{bmatrix}, \\
\mathcal{C}_t &= \begin{bmatrix} CG^{t-1}(B - KD) & CG^{t-3}(B - KD) & \ldots & C(B - KD) \end{bmatrix}.
\end{aligned}
\tag{5}
$$

where $G = A - KC$ is called the closed-loop matrix. The Kalman predictor (3) can be written as

$$
m_{t+1} = \mathcal{O}_t y_{1:t} + \mathcal{C}_t x_{1:t} + Dx_{t+1},
\tag{6}
$$

The prediction error $e_t = y_t - m_t$, also called the *innovation*, is zero-mean with a stationary covariance $V$. Our goal is to design an algorithm $\hat{m}_t(y_{1:t-1}, x_{1:t})$ such that the following regret

$$
\text{Regret}(T) \triangleq \sum_{t=1}^T \|y_t - \hat{m}_t\|_2^2 - \|y_t - m_t\|_2^2
\tag{7}
$$

is bounded by $\text{polylog}(T)$ with high probability.

**Improper learning.** The standard objective (such as squared loss) for learning the parameters of a Kalman predictive model is non-convex as apparent in (3). However, we aspire to an algorithm that optimizes a convex objective for which theoretical guarantees of convergence and sample complexity analysis are possible. This motivates developing an algorithm based on *improper learning*. Instead of directly learning the model parameters in a hypothesis class $\mathcal{H}$ (such as $\mathcal{H} = (A, B, C, D, Q, R)$ in an LDS), improper learning methods reparameterize and learn over a different class $\widetilde{\mathcal{H}}$. The class $\widetilde{\mathcal{H}}$ is often a *relaxation*: it is chosen in a way that is easier to optimize and more computationally efficient while being close to the original hypothesis class. Improper learning has been used to circumvent the proper learning lower bounds [14] and is also deployed in [22, 23, 32, 48].

We develop an algorithm based on improper learning and a tight *convex relaxation*: we slightly overparameterize the LDS predictive model such that the resulting objective is convex. Designing an overparameterized model requires care as too few parameters may result in a large bias whereas too many parameters may result in high variance. Section 5.3 presents our overparameterization approach based on spectral methods that enjoys a small approximation error with relatively few parameters.

# 4 SLIP: Spectral LDS improper predictor

Algorithm 1 presents a pseudocode for the SLIP algorithm. SLIP is based on an online regularized least squares and a linear predictor $\hat{m}_t = \hat{\Theta}^{(t)} f_t$, where $f_t \in \mathbb{R}^l$ is the feature vector and $\hat{\Theta}^{(t)} \in \mathbb{R}^{m \times l}$ is the parameter matrix. The features are constructed from past observations and inputs using eigenvectors of a particular $T \times T$ Hankel matrix with entries

$$H_{ij} = \frac{1 + (-1)^{i+j}}{2(i + j - 1)}, \quad 1 \le i, j \le T. \tag{8}$$

Let $\phi_1, \ldots, \phi_k$ for $k \le T$ be the top $k$ eigenvectors of matrix $H$, to which we refer as *spectral filters*. At every time step, we obtain our feature vector by concatenating the current input $x_t$ to $k$ *output features* based on $y_{1:t-1}$ and $k$ *input features* based on $x_{1:t-1}$. More specifically, we have

$$\begin{aligned} \widetilde{y}_{t-1}(j) &\triangleq (\phi_j^\top(t-1:1) \otimes I_m) y_{1:t-1} = \phi_j(1) y_{t-1} + \cdots + \phi_j(t-1) y_1 \quad \text{(output features),} \\ \widetilde{x}_{t-1}(j) &\triangleq (\phi_j^\top(t-1:1) \otimes I_n) x_{1:t-1} = \phi_j(1) x_{t-1} + \cdots + \phi_j(t-1) x_1 \quad \text{(input features),} \end{aligned} \tag{9}$$

for $j \in \{1, \ldots, k\}$, resulting in a feature vector $f_t$ with dimension $l = mk + nk + n$. Upon receiving a new observation, the parameter matrix is updated by minimizing the regularized loss $\sum_{i=1}^t \|\hat{\Theta} f_t - y_t\|^2 + \alpha \|\hat{\Theta}\|_2^2$ for $\alpha > 0$, which yields the following update rule

$$\hat{\Theta}^{(t+1)} = \left( \sum_{i=1}^t y_i f_i^\top \right) \left( \sum_{i=1}^t f_i f_i^\top + \alpha I_l \right)^{-1}. \tag{10}$$

Importantly, Algorithm 1 requires no knowledge of system parameters, noise covariance, or state dimension and the predictive model is learned online only through sequences of inputs and observations. Note that the spectral filters are computed by conducting a single eigendecomposition and are fixed throughout the algorithm; matrix $\Psi_t$ merely selects certain elements of the spectral filters used for constructing features.

The next theorem analyzes the regret achieved by the SLIP algorithm. A proof sketch of the theorem is provided in Section 6 and a complete proof is deferred to Appendix F.

**Theorem 1. (Regret of the SLIP algorithm)** *Consider system* (4) *without inputs with initial state covariance equal to $P$. Let $m_t$ be the predictions made by the best linear predictor (Kalman filter) and $\hat{m}_t$ be the predictions made by Algorithm 1. Fix the failure probability $\delta > 0$ and assume:*

(i) *There exists a finite $R_\Theta$ that $\|C\|_2, \|P\|_2, \|Q\|_2, \|R\|_2, \|V\|_2 \le R_\Theta$ and $\|\mathcal{O}_t\|_2 \le R_\Theta t^\beta$ for a bounded constant $\beta \ge 0$. Let $\kappa$ be the maximum condition number of $R$ and $Q$.*

(ii) *The system is marginally stable with $\rho(A) \le 1$ and $\|A^t\|_2 \le \gamma t^{\log(\gamma)}$ for a bounded constant $\gamma \ge 1$. Furthermore, the closed-loop matrix $G$ is diagonalizable with real eigenvalues.*

---

**Algorithm 1** SLIP: **S**pectral **LDS I**mproper **P**redictor

---

**Inputs:** Horizon $T$, number of filters $k$, regularization parameter $\alpha$, dimensions $m$ and $n$
**Output:** One-step-ahead predictions $\hat{m}_t(x_{1:t}, y_{1:t-1})$.

Compute the top $k$ eigenvectors $\{\phi_j\}_{j=1}^k$ of matrix $H$ defined in (8).
Set vectors $\psi_i = [\phi_1(i), \ldots, \phi_k(i)]^\top$ for $i \in \{1, \ldots, T\}$, where $\phi_j(i)$ is the $i$-th element of $\phi_j$.
Initialize $\hat{\Theta}^{(1)} \in \mathbb{R}^{m \times l}$ with $l = (n + m)k + n$.
**for** $t = 1, \ldots, T$ **do**
    Set $\Psi_{t-1} = [\psi_{t-1}, \ldots, \psi_1]$, where $\Psi_0 = 0_k$, and compute the feature vector $f_t$:

$$f_t = \begin{bmatrix} \widetilde{y}_{t-1} \\ \widetilde{x}_{t-1} \\ x_t \end{bmatrix} = \begin{bmatrix} (\Psi_{t-1} \otimes I_m) y_{1:t-1} \\ (\Psi_{t-1} \otimes I_n) x_{1:t-1} \\ x_t \end{bmatrix}.$$

    Predict $\hat{m}_t = \hat{\Theta}^{(t)} f_t$.
    Observe $y_t$ and update parameters $\hat{\Theta}^{(t+1)} = \left( \sum_{i=1}^t y_i f_i^\top \right) \left( \sum_{i=1}^t f_i f_i^\top + \alpha I_l \right)^{-1}$.

---

(iii) *The regularization parameter $\alpha$ and the number of filters $k$ satisfy $\alpha \asymp (R_\Theta k T^\beta)^{-1}$ and $k \asymp \log^2(T) \operatorname{polylog}(m, \gamma, R_\Theta, 1/\delta)$.*

(iv) *There exists $s \lesssim_{R_\Theta, m, \gamma, \beta, \delta} t/(k \log k)$ and $t_0$ such that for all $t \geq t_0$*

$$t\Omega_{s/2}(A; \psi) - \Omega_{t+1}(A; \psi) \succeq 0. \tag{11}$$

$\Omega_t(A; \psi)$ *is called the* filter quadratic function *of $\psi$ with respect to $A$ and is defined as*

$$\Omega_t(A; \psi) = (\psi_1^{(d)})(\psi_1^{(d)})^\top + \cdots + (\psi_{t-1}^{(d)} + \cdots + \psi_1^{(d)} A^{t-2})(\psi_{t-1}^{(d)} + \cdots + \psi_1^{(d)} A^{t-2})^\top,$$

*where $\psi_i^{(d)} = [\phi_1(i), \ldots, \phi_k(i)]^\top \otimes I_d$.*

*Then, for all $T \geq \max\{10, t_0\}$, the following holds with probability at least $1 - \delta$,*

$$\operatorname{Regret}(T) \leq \operatorname{polylog}(T, \gamma, \frac{1}{\delta})\kappa \operatorname{poly}(R_\Theta, \beta, m).$$

**Remark 1.** Note that for any matrix $A$, there exists a constant $\gamma \geq 1$ such that $\|A^t\|_2 \leq \gamma t^{\log(\gamma)}$ [27]. We justify our assumption on diagonalizable $G$ with real eigenvalues in the following section. The filter quadratic condition is easily verified for $s > 2(k+1)$ and $t_0 \gtrsim_{R_\Theta, m, \gamma, \beta, \delta} k^2 \log(k)$ for all $A$ with $\rho(A) \leq 1$ for the filters corresponding to truncated observations (a.k.a. basis vectors) such as in [48]. When $A$ is symmetric, this condition can be further simplified to $t\Omega_{s/2}(D; \psi) - \Omega_{t+1}(D; \psi) \succeq 0$ for all diagonal matrices $D$ with $|D_{ii}| \leq 1$.

# 5 Approximation error: Generalized Kolmogorov width

## 5.1 Width of a subset

We now introduce a generalization of *Kolmogorov $k$-width of a subset*, which we later use as a criterion to assess the quality of a function approximation method.

**Definition** 1. **(Generalized Kolmogorov $k$-width)** *Let $W$ be a subset in a normed linear space with norm $\|.\|$ whose elements are $d \times n$ matrices. Given $d \times n$ matrices $u_1, \ldots, u_k$ for $k \geq 1$, let*

$$U(u_1, \ldots, u_k) \triangleq \Big\{ y \ \Big| \ y = \sum\nolimits_{i=1}^k a_i u_i, \ \forall a_i \in \mathbb{R}^{d \times d} \Big\}.$$

*For fixed $k \geq 1$, let $\mathcal{U}_k \triangleq \big\{ U(u_1, \ldots, u_k) \big| u_i \in \mathbb{R}^{d \times n} \big\}$. The generalized $k$-width of $W$ is defined as*

$$d_k(W) \triangleq \inf_{U \in \mathcal{U}_k} \sup_{x \in W} dist(x; U) = \inf_{U \in \mathcal{U}_k} \sup_{x \in W} \inf_{y \in U} \|x - y\|,$$

*where $dist(x; U)$ is the distance of $x$ to subset $U$ and the first infimum is taken over all $U \in \mathcal{U}_k$.*

In words, we are interested in approximating $W$ with the "best" subset in the set $\mathcal{U}_k$: the subset that would minimize the *worst case* projection error of $x \in W$ among all subsets in $\mathcal{U}_k$. This minimal error is given by the generalized $k$-width of $W$. The concept of $k$-width is illustrated in Figure 1.

Definition 1 generalizes the standard Kolmogorov width definition in two ways. First, in our definition $W$, is a subset of matrices whereas, in the original Kolmogorov width, $W$ is a subset of vectors. This generalization is necessary as we wish to approximate the coefficient set of the Kalman predictive model whose elements $\mathcal{O}_t$ and $\mathcal{C}_t$ are matrices. Second, we allow the coefficients $a_i$ to be matrices, generalizing over the scalar coefficients used in the original Kolmogorov width. Allowing coefficients to be matrices gives flexibility for finding a reparameterization with a small approximation error.

## 5.2 From a small width to an efficient convex relaxation

Before stating our approximation technique, we briefly describe how a small generalized $k$-width can allow for an efficient convex relaxation.

To understand the main idea, consider system (4) with no inputs whose predictive model can be written as $m_{t+1} = \mathcal{O}_t y_{1:t}$. Matrix $\mathcal{O}_t$ belongs to a subset in $\mathbb{R}^{m \times mt}$ restricted by the constraints on system parameters. A naive approach for a convex relaxation is learning $\mathcal{O}_t$ directly. However in this approach, the total number of parameters is $m^2 t$, which hinders achieving sub-linear regret.

Now suppose that there exists $k \ll t$ for which the generalized $k$-width is small, i.e. there exist *fixed known* matrices $u_1, \ldots, u_k \in \mathbb{R}^{m \times mt}$ that approximate any $\mathcal{O}_t$ with a small error $\mathcal{O}_t \approx \sum_{i=1}^{k} a_i u_i$, where $a_1, \ldots, a_k \in \mathbb{R}^{m \times m}$ are coefficient matrices. The predictive model can be approximated by $m_{t+1} \approx \sum_{i=1}^{k} a_i u_i y_{1:t}$, provided that the norm of $y_{1:t}$ (compared to the approximation error of $\mathcal{O}_t$) is controlled with high probability. Since $u_i$ are known, we only need to learn coefficients $a_i$ resulting in a total of $m^2 k$ parameters which is much smaller than the naive approach with $m^2 t$ parameters.

## 5.3 Filter approximation

Consider the matrix $\mu(G) \triangleq [I, G, G^2, \ldots, G^{T-1}]$, where $G \in \mathbb{R}^{d \times d}$ is a real square matrix with spectral radius $\rho(G) \leq 1$. We seek to approximate $\mu(G) \approx \widetilde{\mu}(G) = \sum_{i=1}^{k} a_i u_i$ by a linear combination of $k$ matrices $u_1, \ldots, u_k \in \mathbb{R}^{d \times Td}$ and coefficient matrices $a_1, \ldots, a_k \in \mathbb{R}^{d \times d}$. We evaluate the quality of approximation in operator 2-norm $\|\mu(G) - \widetilde{\mu}(G)\|_2$ by studying the generalized $k$-width of $\mu(G)$.

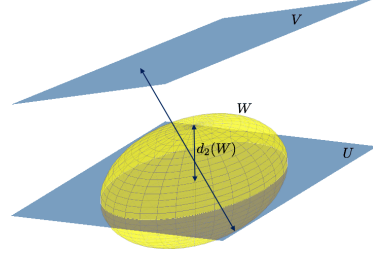

Figure 1: Approximating a 3D ellipsoid $W$ by a 2D plane $U(u_1, u_2)$ among $\mathcal{U}_2$, the set of all planes. Here, $U$ has the smallest worst-case projection error that is equal to the 2-width of $W$ denoted by $d_2(W)$.

We demonstrate a sharp phase transition. Precisely, we show that when $G$ is diagonalizable with real eigenvalues, the width $d_k(W)$ decays exponentially fast with $k$, but for a general $G$ with $d \geq 2$ it decays only polynomially fast. In other words, when $d \geq 2$ the inherent structure of the set $W$ is not easily exploited by linear subspaces.

**Theorem 2.** **(Kalman filter $k$-width)** *Let* $W \triangleq \{[I, G, G^2, \ldots, G^{T-1}] \mid \rho(G) \leq 1, G \in \mathbb{R}^{d \times d}\}$ *and endow the space of $W$ with the 2-norm. The following bounds hold on the $k$-width of the set $W$:*

1. *If $d \geq 2$, then for $1 \leq k \leq T$, we have $d_k(W) \geq \sqrt{T-k}$.*

2. *Restrict $G$ to be diagonalizable with real eigenvalues. If $T \geq 10$, then for any $d \geq 1$*

$$d_k(W) \leq C_0 d\sqrt{T}(\log T)^{1/4} c^{-k/\log T},$$

*where $c = \exp(\pi^2/16)$ and $C_0 = \sqrt{43}$. Moreover, there exists an efficient spectral method to compute a $k$-dimensional subspace that satisfies this upper bound.*

*Proof.* We only provide a proof sketch for the second claim; see Appendix C for a complete proof. Let $\lambda_1, \ldots \lambda_d \in [-1, 1]$ be the eigenvalues of $G$. Let $v_i$ be the right eigenvectors of $G$ and $w_i^\top$ be the left eigenvectors of $G$ and write

$$\mu(G) = \sum_{i=1}^{d} v_i w_i^\top ([1, \lambda_i, \ldots, \lambda_i^{T-1}] \otimes I_d) = \sum_{i=1}^{d} v_i w_i^\top (\mu(\lambda_i) \otimes I_d).$$

We approximate the row vector $\mu(\lambda)$ for any $\lambda \in [-1, 1]$ using principal component analysis (PCA). The covariance matrix of $\mu(\lambda)$ with respect to a uniform measure is given by

$$H = \int_{\lambda=-1}^{1} \frac{1}{2} \mu(\lambda)^\top \mu(\lambda) d\lambda \quad \Rightarrow \quad H_{ij} = \int_{-1}^{1} \frac{1}{2} \lambda^{i-1} \lambda^{j-1} d\lambda = \frac{(-1)^{i+j} + 1}{2(i+j-1)}.$$

Let $\{\phi_j\}_{j=1}^{k}$ be the top $k$ eigenvectors of $H$. We approximate $\mu(\lambda)$ by $\widetilde{\mu}(\lambda) = \sum_{j=1}^{k} \langle \mu^\top(\lambda), \phi_j \rangle \phi_j^\top$:

$$\mu(G) \approx \widetilde{\mu}(G) = \sum_{j=1}^{k} \Big[ \sum_{i=1}^{d} \langle \mu^\top(\lambda_i), \phi_j \rangle v_i w_i^\top \Big] (\phi_j^\top \otimes I_d) = \sum_{j=1}^{k} a_j u_j.$$

We show a uniform bound on $\|\mu(G) - \widetilde{\mu}(G)\|$ by first analyzing the PCA approximation error which depends on the spectrum of $H$. Matrix $H$ is a positive semi-definite (PSD) Hankel matrix, a square matrix whose $ij$-th entry only depends on the sum $i + j$. We leverage a recent result by Beckermann and Townsend [3] who proved that the spectrum of PSD Hankel matrices decays exponentially fast. This result, however, only guarantees a small *average* error but we need to prove that the *maximum* error is small to ensure a uniform bound on regret. Observe that the PCA error $r(\lambda) = \mu(\lambda) - \widetilde{\mu}(\lambda)$ is defined over a finite interval $[-1, 1]$ with a small average. By computing the Lipschitz constant of $r(\lambda)$, we show that the maximum PCA error is small, resulting in an upper bound on $d_k(W)$. ∎

The approximation technique used in the above theorem can readily be applied to approximate the Kalman predictive model by $m_t \approx \widetilde{m}_t \triangleq \widetilde{\mathcal{O}}_t y_{1:t-1} + \widetilde{\mathcal{C}}_t x_{1:t-1} + D x_t$. A complete derivation of convex relaxation along with an approximation error analysis is provided in Appendix D.

## 6 Proof roadmap of Theorem 1

In this section we present a proof sketch for Theorem 1; a complete proof is deferred to Appendix E and Appendix F. Let $e_t = y_t - m_t$ denote the innovation process, $b_t = \widetilde{m}_t - m_t$ denote the bias due to convex relaxation, and $\mathcal{L}(T) \triangleq \sum_{t=1}^T \|\hat{m}_t - m_t\|_2^2$ measure the difference between SLIP's predictions and the Kalman predictions in hindsight. Regret defined in (7) can be written as

$$\mathrm{Regret}(T) = \sum_{t=1}^T \|\hat{m}_t - m_t\|_2^2 - \sum_{t=1}^T 2 e_t^\top (\hat{m}_t - m_t) = \mathcal{L}(T) - \sum_{t=1}^T 2 e_t^\top (\hat{m}_t - m_t). \quad (12)$$

Using an argument based on self-normalizing martingales, the second term is shown to be of order $\sqrt{\mathcal{L}(T)}$ and thus, it suffices to establish a bound on $\mathcal{L}(T)$. Define $Z_t \triangleq \alpha I + \sum_{i=1}^t f_i f_i^\top, E_t \triangleq \sum_{i=1}^t e_i f_i^\top$, and $B_t \triangleq \sum_{i=1}^t b_i f_i^\top$. A straighforward decomposition of loss gives

$$\mathcal{L}(T) \le 3 \underbrace{\sum_{i=1}^T \|E_{t-1} Z_{t-1}^{-1} f_t\|_2^2}_{\text{least squares error}} + 3 \underbrace{\sum_{i=1}^T \|B_{t-1} Z_{t-1}^{-1} f_t + b_t\|_2^2}_{\text{improper learning bias}} + 3 \underbrace{\sum_{i=1}^T \|\alpha \tilde{\Theta} Z_{t-1}^{-1} f_t\|_2^2}_{\text{regularization error}}. \quad (13)$$

Among all, it is most difficult to establish a bound on the least squares error. We restrict our attention to this term and refer to appendix for bounds on improper learning bias and regularization error.

**Least squares error.** Consider the following upper bound

$$\sum_{t=1}^T \|E_{t-1} Z_{t-1}^{-1} f_t\|_2 \le \max_{1 \le t \le T} \|E_{t-1} Z_{t-1}^{-1/2}\|_2 \sum_{t=1}^T \|Z_{t-1}^{-1/2} f_t\|_2.$$

We show the first term is bounded by $\mathrm{polylog}(T)$ for any $\delta \ge 0$. In particular,

$$\max_{1 \le t \le T} \|E_{t-1} Z_{t-1}^{-1/2}\|_2 \lesssim_{R_\Theta, m, \gamma, \beta, \delta} \max_{1 \le t \le T} \log\left( \det(Z_t) \det(\alpha I)^{-1} \delta^{-1} \right) \lesssim_{R_\Theta, m, \gamma, \beta, \delta} k \log(T).$$

Our argument is based on vector self-normalizing martingales, a similar technique used in [1, 43, 48]. $\det(Z_t)$ is bounded by $\mathrm{poly}(T)$ because (1) the feature dimension is $\mathrm{polylog}(T)$ on account of Theorem 2 and (2) $\rho(A) \le 1$ ensures that features and thus $Z_t$ grow at most polynomially in $t$.

We use an argument inspired by Lemma 2 in [30] and Schur complements [56] to conclude that

$$\sum_{t=1}^T \|Z_{t-1}^{-1/2} f_t\|_2^2 \asymp_M \mathrm{polylog}(T) \quad \Leftrightarrow \quad Z_{t-1} - \frac{1}{c_T} f_t f_t^\top \succeq 0 \quad \text{for} \quad c_T \asymp_M \mathrm{polylog}(T).$$

We show a high probability Löwner upper bound on $f_t f_t^\top$ based on $\mathrm{cov}(f_t)$ using sub-Gaussian quadratic tail bounds [50]. To capture the excitation behavior of features, we establish a Löwner lower bound on $Z_t$ by proving that the process $\{f_t\}_{t \ge 1}$ satisfies a *martingale small-ball condition* [38, 46]. We leverage the small-ball condition lower tail bounds and prove the following lemma.

**Lemma 1. (Martingale small-ball condition)** *Let $\phi_1, \ldots, \phi_k \in \mathbb{R}^T$ be orthonormal and fix $\delta > 0$. Given system* (4)*, let $\mathcal{F}_t = \sigma\{\eta_0, \ldots, \eta_{t-1}, \zeta_1, \ldots, \zeta_t\}$ be a filteration and for all $t \ge 1$ define $f_t = \psi_1 \otimes y_{t-1} + \cdots + \psi_{t-1} \otimes y_1$, where $\psi_i = [\phi_1(i), \ldots, \phi_k(i)]^\top$.*

*1. Let $\Gamma_i = \mathrm{cov}(f_{t+i}|\mathcal{F}_t)$. For $1 \le s \le T$, the process $\{f_t\}_{t \ge 1}$ satisfies a $(s, \Gamma_{s/2}, p = 3/20)$-block martingale small-ball (BMSB) condition, i.e. for any $t \ge 0$ and fixed $\omega$ in unit sphere $\mathcal{S}^{l-1}$*

$$\frac{1}{s} \sum_{i=1}^s \mathbb{P}\left( |\omega^\top f_{t+i}| \ge \sqrt{\omega^\top \Gamma_{s/2} \omega} \mid \mathcal{F}_t \right) \ge p.$$

*2. Under the assumptions of Theorem 1, the following holds with probability at least $1 - \delta$*

$$\sum_{t=1}^T \|Z_{t-1}^{-1/2} f_t\|_2^2 \le \kappa k^2 \log(T) \mathrm{poly}\left(R_\Theta, \beta, m, \log(\gamma), \log\left(\frac{1}{\delta}\right)\right).$$

**Remark 2.** While the algorithm derivation, convex relaxation approximation error, and most of the regret analysis consider a system with control inputs, the excitation result of Lemma 1 is given without inputs. We believe that extending our analysis for LDS with inputs is possible by characterizing the input features and in light of the experiments. However, such an extension requires some care. For instance, one needs to characterize the covariance between features constructed from observations and features constructed from inputs to demonstrate a small-ball condition.

## 7 Experiments

We carry out experiments to evaluate the empirical performance of our provable method in three dynamical systems with long-term memory. We compare our results against those yielded by the wave filtering algorithm [22] implemented with follow the regularized leader and the truncated filtering algorithm [48]. We consider $\|\hat{m}_t - m_t\|^2$, the squared error between algorithms predictions and predictions by a Kalman filtering algorithm that knows system parameters, as a performance measure. For all algorithms, we use $k = 20$ filters and run each experiment independently 100 times and present the average error with 99% confidence intervals.

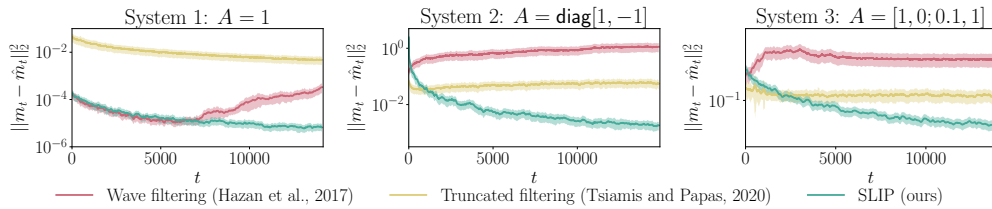

Figure 2: Performance of our algorithm compared with wave filtering and truncated filtering. System 1 is an scalar LDS with $A = B = D = 1$, $C = Q = R = 0.001$, and $x_t \sim \mathcal{N}(0, 2)$. System 2 is a multi-dimensional LDS with no inputs and $A = \text{diag}[-1, 1]$, $C = [0.1, 0.5]$, $R = 0.5$, and $Q = [4, 6; 6, 10] \times 10^{-3}$. System 3 is another multi-dimensional LDS with $A = [1, 0; 0.1, 1]$, $x_i \sim \mathcal{U}(-0.01, 0.01)$, $Q = 10^{-3}I$, $R = I$, $C = [0, 0.1; 0.1, 1]$, and $B, D$ are matrices of all ones.

In the first example (Figure 2, left), we consider a scalar marginally stable system with $A = 1$ and Gaussian inputs. This system exhibits long forecast memory with $G \approx 0.999$. Observe that the truncated filter suffers from a large error which is due to ignoring long-term dependencies. The wave filter predictions also deviates from optimal predictions as it only considers $y_{t-1}, x_{1:t}$ for predicting $y_t$. The middle plot in Figure 2 presents the results for a multi-dimensional system with $A = \text{diag}[-1, 1]$ and no inputs. This system also has a long forecast memory ($G$ has eigenvalues $\approx \{0.991, -0.932\}$), resulting in poor performance of the truncated filter. The wave filter also performs poorly in this system as it is only driven by stochastic noise. For the last example, we consider another multi-dimensional system where $A$ is a lower triangular matrix (Figure 2, right). This is a difficult example where $\rho(A) = 1$ but $\|A\|_2 > 1$, resulting in a polynomial growth of the observations over time. The results show that our algorithm outperforms both the wave filter, which requires a symmetric $A$, and the truncated filter in the case of fast-growing observations.

Experiments on the hyperparameter sensitivity of our algorithm and comparison with the EM algorithm are provided in Appendix H.

## 8 Discussion and future work

We presented the SLIP algorithm, an efficient algorithm for learning a predictive model of an unknown LDS. Our algorithm provably and empirically converges to the optimal predictions of the Kalman filter given the true system parameters, even in the presence of long forecast memory. We analyzed the generalized $k$-width of the Kalman filter coefficient set with the closed-loop matrix $G$ and obtained a low-dimensional linear approximation of the Kalman filter when $G$ is diagonalizable with real eigenvalues. We proved that without assuming real eigenvalues, the Kalman filter coefficient set is difficult to approximate by linear subspaces. Our approach of studying $k$-width as a measure for the possibility of an efficient convex relaxation may be of independent interest. Important future directions are to design efficient algorithms that handle arbitrary $G$ and to provide theoretically guaranteed uncertainty estimation for prediction.

## 9 Broader impact

Because linear dynamical systems are a fundamental tool in essentially all quantitative disciplines (engineering, physical sciences, life sciences, social sciences), advances in the capabilities for learning and predicting such systems may have very significant positive consequences. (For example, currently Google Scholar lists 912,000 papers that mention Kalman filter.) Our proposed algorithm is practical, fast, easy to implement, and provably more robust to a wider range of conditions than previous algorithms. In particular, many real-world systems exhibit long-term memory and a wide range of time scales, which our approach handles well.

Like any very general computational tool, the algorithm can be applied in contexts where the societal consequences may be negative. To our knowledge, the vast majority of uses for linear dynamical systems involve human experts studying and predicting systems of interest, such as climate systems or ecologies. In these contexts the effects of improved prediction and reliability would typically be positive. It is specifically unlikely that LDS would be used to model individual humans algorithmically, since humans are decidedly not linear systems.

## Acknowledgements

The authors would like to thank the anonymous reviewers for their comments and suggestions, which helped improve the quality and clarity of the manuscript. This work is supported by the Scalable Collaborative Human-Robot Learning (SCHooL) Project, an NSF National Robotics Initiative Award 1734633. The work of Jiantao Jiao was partially supported by NSF Grants IIS-1901252 and CCF-1909499.

## Footnotes

[1]One can interpret the Kalman filter Equation (1) as linear combinations of optimal predictor given existing data $A\hat{h}_{t|t-1}$, known drift $Bx_t$, and amplified innovation $K(y_t - C\hat{h}_{t|t-1} - Dx_t)$, where the term $y_t - C\hat{h}_{t|t-1} - Dx_t$, called the *innovation* of process $y_t$, measures how much additional information $y_t$ brings compared to the known information of observations up to $y_{t-1}$.

[2]Indeed, a common practice is to set the timescale to be small enough to handle the fastest-changing variables.

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
