[Supplementary Material]

# Supplementary Material for "SLIP: Learning to Predict in Unknown Dynamical Systems with Long-Term Memory"

**Paria Rashidinejad**     **Jiantao Jiao**     **Stuart Russell**
EECS Department, University of California, Berkeley
{paria.rashidinejad, jiantao, russell}@berkeley.edu

## Contents

# Guide to the appendix

The appendix is organized as follows.

In Appendix A, we present a matrix representation of system (4) describing aggregated observations $y_{1:t}$ in terms of past inputs and noise. We also restate our matrix representation of the Kalman predictive model.

In Appendix B, we provide upper bounds on the matrix coefficients used in the aggregated system representation as well as a high probability upper bound on the norm of observations $\|y_{1:t}\|_2$. We also discuss our assumption on the 2-norm of the Kalman coefficient matrices (control matrix $\mathcal{C}_t$ and observation matrix $\mathcal{O}_t$) and present two examples providing bounds on the 2-norm of these coefficients.

In Appendix C, we first analyze the error of approximating $\mu(\lambda)$ by spectral methods, considering the spectrum of the Hankel covariance matrix. A proof of Theorem 2 is presented in Appendix C.2.

In Appendix D, we analyze convex relaxation approximation error and show that the convex relaxation bias is small with high probability, provided that the number of filters $k \gtrsim_M \log^2(T)$.

In Appendix E, we write a bound on regret decomposed into least squares error, improper learning bias, regularization error, and innovation error. We further extract the term $\|Z_{t-1}^{-1/2} f_t\|_2^2$ making the bound ready for analysis in subsequent sections.

In Appendix F, we provide our regret analysis. In Appendix F.1, we present a high probability bound on $\det(Z_t)$ that appears multiple times throughout our analysis. In Appendix F.2, we derive a result on self-normalizing vector martingales that assists bounding several terms. In Appendix F.3, we provide a bound on $\|Z_{t-1}^{-1/2} f_t\|_2^2$ using sub-Gaussian tail properties, a block-martingale small-ball condition, and a filter quadratic function condition. The proof of Lemma 1 is given in Appendix F.4. The regularization term and innovation error are analyzed in Appendix F.5 and Appendix F.6, respectively. The proof of the regret theorem is presented in Appendix F.7.

A few technical lemmas are presented in Appendix G. Additional experiments are presented in Appendix H. In Appendix I, a discussion on systems with long forecast memory is provided.

# A  Aggregated representations

We start by introducing an aggregated notation for representing linear dynamical systems and the Kalman predictive model.

## A.1  Linear dynamical systems

For the linear dynamical system of (4), define the following matrices

$$
\mathcal{T}_t = \begin{bmatrix} C & 0 & 0 & \ldots & 0 \\ CA & C & 0 & \ldots & 0 \\ CA^2 & CA & C & \ldots & 0 \\ \vdots & \vdots & \vdots & \ddots & \vdots \\ CA^{t-1} & CA^{t-2} & CA^{t-3} & \ldots & C \end{bmatrix} \begin{bmatrix} AP^{1/2} & 0 & 0 & \ldots & 0 \\ 0 & Q^{1/2} & 0 & \ldots & 0 \\ 0 & 0 & Q^{1/2} & \ldots & 0 \\ \vdots & \vdots & \vdots & \ddots & \vdots \\ 0 & 0 & 0 & \ldots & Q^{1/2} \end{bmatrix},
$$

$$
\mathcal{I}_t = \begin{bmatrix} D & 0 & 0 & \ldots & 0 \\ CB & D & 0 & \ldots & 0 \\ CAB & CB & D & \ldots & 0 \\ \vdots & \vdots & \vdots & \ddots & \vdots \\ CA^{t-2}B & CA^{t-3}B & CA^{t-4}B & \ldots & D \end{bmatrix}, \tag{14}
$$

$$
\mathcal{R}_t = \begin{bmatrix} R^{1/2} & 0 & 0 & \ldots & 0 \\ 0 & R^{1/2} & 0 & \ldots & 0 \\ 0 & 0 & R^{1/2} & \ldots & 0 \\ \vdots & \vdots & \vdots & \ddots & \vdots \\ 0 & 0 & 0 & \ldots & R^{1/2} \end{bmatrix}.
$$

Let $\mathcal{K}_t \mathcal{K}_t^\top = \mathcal{T}_t \mathcal{T}_t^\top + \mathcal{R}_t \mathcal{R}_t^\top$, where $\mathcal{K}_t$ is the unique solution to Cholesky decomposition. The system observations $y_{1:t}$ can be written as

$$
y_{1:t} = \mathcal{K}_t \xi_{1:t} + \mathcal{I}_t x_{1:t}, \tag{15}
$$

where $\xi_i \in \mathbb{R}^m$ is a Gaussian random vector with covariance $I_m$.

## A.2  Kalman filter

For convenience, we restate our notation of the Kalman predictive model from Section 3. Define the following matrices

$$
\begin{aligned}
\mathcal{O}_t &= \begin{bmatrix} CG^{t-1}K & CG^{t-3}K & \ldots & CK \end{bmatrix}, \\
\mathcal{C}_t &= \begin{bmatrix} CG^{t-1}(B - KD) & CG^{t-2}(B - KD) & \ldots & C(B - KD) \end{bmatrix}.
\end{aligned} \tag{16}
$$

We refer to $\mathcal{O}_t$ and $\mathcal{C}_t$ as *observation matrix* and *control matrix*, respectively. Using the above notation, the Kalman prediction $m_{t+1}$ is given by

$$
m_{t+1} = \mathcal{O}_t y_{1:t} + \mathcal{C}_t x_{1:t} + D x_{t+1}.
$$

# B  Norm bounds

As a preliminary step, we compute a few bounds that will be used later in the regret analysis of the SLIP algorithm. In particular, we compute upper bounds on the norms of parameter matrices defined in (14) and discuss upper bounds on the norms of observation and control matrix of the Kalman predictive model. Further, we derive a high probability upper bound on the observation norm.

## B.1  Bounds on parameters

The following lemma provides upper bounds on the norm of matrices that describe a linear dynamic system.

**Lemma** B.1. **(LDS parameter bounds)** *Consider system (4). Let $R_P = \max\{\|B\|_2, \|C\|_2, \|D\|_2\}$ and $R_C = \max\{\|P\|_2, \|Q\|_2, \|R\|_2\}$. Suppose that $\|A^t\|_2 \leq \gamma t^{\log(\gamma)}$ for a bounded constant $\gamma \geq 1$. For $\mathcal{T}_t$, $\mathcal{I}_t$, and $\mathcal{K}_t$ defined in (14), the following operator norm bounds hold:*

(i) $\|\mathcal{T}_t\|_2 \leq R_C^{1/2} R_P \gamma (1 + \gamma) t^{\log(\gamma)+1}$,

(ii) $\|\mathcal{I}_t\|_2 \leq R_P[1 + t\gamma t^{\log(\gamma)}]$,

(iii) $\|\mathcal{K}_t\|_2 \leq \sqrt{R_C + R_C R_P^2 (1 + \gamma)^4 t^{2\log(\gamma)+2}}$.

*Proof.* By Lemma G.1,

$$\|\mathcal{T}_t\|_2 \leq (\|A\|_2 + 1) R_C^{1/2} \|C\|_2 \sum_{i=1}^{t} \|A^i\|_2 \leq R_C^{1/2} R_P \gamma (1 + \gamma) t^{\log(\gamma)+1}.$$

Similarly,

$$\|\mathcal{I}_t\|_2 \leq \|D\|_2 + \|C\|_2 \|B\|_2 \sum_{i=1}^{t} \|A^i\|_2 \leq R_P + R_P^2 \gamma t^{\log(\gamma)+1}.$$

It follows by the sub-additive property of matrix operator norm that

$$\|\mathcal{K}_t \mathcal{K}_t^\top\|_2 = \|\mathcal{K}_t\|_2^2 \leq \|\mathcal{T}_t\|_2^2 + \|\mathcal{R}_t\|_2^2 \quad \Rightarrow \quad \|\mathcal{K}_t\|_2 \leq \sqrt{R_C + R_C R_P^2 (1 + \gamma)^4 t^{2\log(\gamma)+2}}.$$

∎

In the regret analysis, we assume that $\|\mathcal{O}_t\|_2 \leq R_\mathcal{O} t^\beta$ for a finite $\beta \geq 0$. We justify this assumption in the examples below. The following example shows that $\beta = 0$ when the system is single-input single-output (SISO).

**Example** B.1. **(Observation matrix norm bound in SISO systems)** *For a SISO linear dynamical system, the following equation holds*

$$KC = \frac{A\Sigma^+ C^2}{\Sigma^+ C^2 + R} \Rightarrow 0 \leq KC \leq A.$$

*We have $G = A - KC$. Applying the above constraint gives*

$$G \leq A$$

*The squared norm of vector $\mathcal{O}_t$ is given by*

$$\|\mathcal{O}_t\|_2^2 = \sum_{i=0}^{t-1} (KCG^i)^2 = \sum_{i=0}^{t-1} (A - G)^2 G^{2i}.$$

*Under the constraint $G \leq A \leq 1$, the maximum of $\|\mathcal{O}_t\|_2^2$ is 1 obtained when $G = 0$ and $A = 1$.*

In the following example, we compute a loose upper bound on $\|\mathcal{O}_t\|_2$.

**Example** B.2. **(Observation matrix norm bound in MIMO systems with d = m)** *Consider an LDS with $d = m$. We begin by computing an upper bound on the norm of the Kalman gain. Let $K = AK'$. By the recursive updates of a stationary Kalman gain, we write*

$$CK' = C\Sigma^+ C^\top [C\Sigma^+ C^\top + Q]^{-1} \preceq I \Rightarrow \|CK'\|_2 \leq 1.$$

*Lower bounding $\|CK\|_2$ yields*

$$\|K'\|_2 \sigma_{\min}(C) \leq \|CK'\|_2 \leq 1 \Rightarrow \|K'\|_2 \leq \frac{1}{\sigma_{\min}(C)}.$$

*Let $\kappa_C = \sigma_{\max}(C)/\sigma_{\min}(C)$ to be the condition number of $C$. Assume $\|G^t\|_2 \leq \gamma_g t^{\log(\gamma_g)}$. We have*

$$\|\mathcal{O}_t\|_2 \leq \sum_{i=1}^{t} \|C\|_2 \|G^i\|_2 \|K'\|_2 \leq \kappa_C \gamma_g t^{\log(\gamma_g)+1}.$$

## B.2 Bound on observation norm

One of the quantities that appear in the regret analysis of our algorithm is the squared norm of $y_{1:t}$. The following lemma provides a high probability upper bound for $\|y_{1:t}\|_2^2$.

**Lemma** B.2. **(Observation norm bound)** *Consider system (4). Let* $R_P = \max\{\|B\|_2, \|C\|_2, \|D\|_2\}$, $R_C = \max\{\|P\|_2, \|Q\|_2, \|R\|_2\}$, *and* $\|x_t\|_2 \leq R_x$. *Suppose that* $\|A^t\|_2 \leq \gamma t^{\log(\gamma)}$ *for a bounded constant* $\gamma \geq 1$. *For any* $\delta > 0$ *and all* $t \geq 0$,

$$\mathbb{P}\left[\|y_{1:t}\|_2^2 \geq 6(R_P^2 + 1)(R_x^2 + R_C)(1 + \gamma)^4(mt + \delta)t^{2+2\log(\gamma)}\right] \leq e^{-\delta}.$$

*Proof.* From (15), we see that

$$\|y_{1:t}\|_2^2 \leq 2\|\mathcal{I}_t\|_2^2\|x_{1:t}\|_2^2 + 2\|\mathcal{K}_t\|_2^2\|\xi_{1:t}\|_2^2$$

Using Gaussian upper tail bounds [5], we have

$$\mathbb{P}\left[\|\xi_{1:t}\|_2^2 > 2mt + 3\delta\right] \leq \mathbb{P}\left[\|\xi_{1:t}\|_2^2 > mt + 2\sqrt{mt\delta} + 2\delta\right] \leq e^{-\delta}.$$

Using the bounds computed in Lemma B.1, the following holds with probability at least $1 - e^{-\delta}$

$$\begin{aligned}
\|y_{1:t}\|_2^2 &\leq 2\|\mathcal{I}_t\|_2^2\|x_{1:t}\|_2^2 + 2\|\mathcal{K}_t\|_2^2\|\xi_{1:t}\|_2^2 \\
&\leq 6(R_P^2 + 1)(R_x^2 + R_C)(1 + \gamma)^4(mt + \delta)t^{2+2\log(\gamma)}.
\end{aligned} \tag{17}$$

∎

## C Filter approximation and width analysis

In this section we first provide a series of lemmas characterizing the reconstruction error of applying PCA to approximate the vector function $\mu(\lambda) = [1, \lambda, \ldots, \lambda^{T-1}]$. These lemmas are later used to prove Theorem 2.

### C.1 Bounds on PCA approximation error

The goal of this section is to establish a uniform bound on the norm of the reconstruction error of approximating $\mu(\lambda)$ with $\widetilde{\mu}(\lambda)$. The following lemma states a standard result on the average PCA reconstruction error, presented here for completeness.

**Lemma** C.1. **(Average reconstruction error bound)** *Let* $\mu(\lambda) \in \mathbb{R}^T$ *be a vector function parameterized by* $\lambda \in \mathcal{A}$. *Define the following matrix with respect to probability measure* $p$

$$Z = \int_{\mathcal{A}} \mu(\lambda)\mu^\top(\lambda)p(d\lambda).$$

*Let* $\{(\sigma_j, \phi_j)\}_{j=1}^T$ *be the eigenpairs of* $Z$. *Let* $\widetilde{\mu}(\lambda)$ *be the projection of* $\mu(\lambda)$ *to the linear subspace spanned by* $\{\phi_1, \ldots, \phi_k\}$. *Then,*

$$\int_{\mathcal{A}} \|\mu(\lambda) - \widetilde{\mu}(\lambda)\|_2^2 p(d\lambda) = \sum_{j=k+1}^T \sigma_j.$$

*Proof.* Define $U_k$ to be a $T \times k$ matrix with columns $\phi_1, \ldots \phi_k$, the eigenvectors of matrix $Z$. The reconstruction error can be written as

$$r(\lambda) = \mu(\lambda) - U_k U_k^\top \mu(\lambda) = (I - U_k U_k^\top)\mu(\lambda) = \Pi_k \mu(\lambda).$$

The average squared norm of reconstruction error is given by

$$\int_{\mathcal{A}} \|r(\lambda)\|_2^2 p(d\lambda) = \int_{\mathcal{A}} \text{tr}[r(\lambda)r(\lambda)^\top]p(d\lambda) = \int_{\mathcal{A}} \text{tr}[\Pi_k \mu(\lambda)\mu(\lambda)^\top \Pi_k^\top]p(d\lambda)$$

$$= \text{tr}[\Pi_k \int_{\mathcal{A}} \mu(\lambda)\mu(\lambda)^\top p(d\lambda)\Pi_k^\top] = \text{tr}[\Pi_k Z \Pi_k^\top] = \sum_{j=k+1}^T \sigma_j.$$

∎

We then use Lipschitz continuity of $\mu(\lambda)$ over the interval $[-1, 1]$ to establish a uniform bound on the reconstruction error.

**Lemma** C.2. *Let $\mu(\lambda) = [1, \lambda, \lambda^2, \ldots, \lambda^{T-1}]^\top$ for $\lambda \in [-1, 1]$ and define*

$$H = \int_{-1}^{1} \frac{1}{2}\mu(\lambda)\mu(\lambda)^\top d\lambda.$$

*Let $\{(\sigma_j, \phi^j)\}_{j=1}^T$ be the eigenpairs of $H$, where $\sigma_j$ are in decreasing order. Let $\widetilde{\mu}(\lambda)$ be the projection of $\mu(\lambda)$ to the linear subspace spanned by $\{\phi_1, \ldots, \phi_k\}$. Then, for any $\lambda \in [-1, 1]$ and $T \geq 1$,*

$$\|\mu(\lambda) - \widetilde{\mu}(\lambda)\|_2^2 \leq T\sqrt{2\sum_{j=k+1}^{T}\sigma_j}.$$

*Proof.* Let us first compute an upper bound on the Lipschitz constant of $\mu(\lambda)$ over $\lambda \in [-1, 1]$. The Lipschitz constant of $\mu(\lambda)$ is bounded by the norm of Jacobian $J(\mu(\lambda)) = [0, 1, 2\lambda, \ldots, (T-1)\lambda^{T-2}]$. Thus,

$$\frac{\|\mu(\lambda_2) - \mu(\lambda_1)\|_2}{|\lambda_2 - \lambda_1|} \leq \|J(\mu(\lambda))\|_2 \leq \sqrt{\sum_{t=1}^{T-1} t^2} \leq \sqrt{T^3/3}.$$

Define $U_k$ to be a matrix with columns $\phi_1, \ldots \phi_k$. The reconstruction error can be written as $r(\lambda) = (I - U_kU_k^\top)\mu(\lambda) = \Pi_k\mu(\lambda)$. A Lipschitz constant for reconstruction error norm is given by

$$
\begin{aligned}
\|r(\lambda_2)\|_2 - \|r(\lambda_1)\|_2 &\leq \|r(\lambda_2) - r(\lambda_1)\|_2 && \text{(inverse triangle inequality)} \\
&= \|\Pi_k(\mu(\lambda_2) - \mu(\lambda_1))\|_2 && \\
&\leq \|\Pi_k\|_2\|(\mu(\lambda_2) - \mu(\lambda_1))\|_2 && \text{(multiplicative property of norm)} \\
&\leq \|(\mu(\lambda_2) - \mu(\lambda_1))\|_2 && (\Pi_k \text{ is contractive}) \\
&\leq \sqrt{T^3/3}|\lambda_2 - \lambda_1| && \text{(Lipschitz continuity of } \mu(\lambda))
\end{aligned}
$$

Thus, an upper bound on the Lipschitz constant of $\|r(\lambda)\|_2^2$ can be computed

$$
\begin{aligned}
\|r(\lambda_2)\|_2^2 - \|r(\lambda_1)\|_2^2 &= (\|r(\lambda_2)\|_2 - \|r(\lambda_1)\|_2)(\|r(\lambda_2)\|_2 + \|r(\lambda_1)\|_2) \\
&\leq \left(\sqrt{T^3/3}|\lambda_2 - \lambda_1|\right)\left(2\max_\lambda\|r(\lambda)\|_2\right) \\
&\leq 2\sqrt{T^3/3}\|\Pi_k\|_2\max_\lambda\|\mu(\lambda)\|_2|\lambda_2 - \lambda_1| \\
&\leq 2T^2|\lambda_2 - \lambda_1|.
\end{aligned}
$$

Let $R_r = \max_\lambda\|r(\lambda)\|_2^2$. On the account of Lemma C.1, $\|r(\lambda)\|_2^2$ has a bounded average over the interval $[-1, 1]$. A bounded and $(2T^2)$-Lipschitz function that achieves the maximum $R_r$ has a triangular shape. It follows that

$$\frac{R_r^2}{2T^2} \geq \sum_{j=k+1}^{T}\sigma_j \quad \Rightarrow \quad \|r(\lambda)\|_2^2 \leq R_r \leq T\sqrt{2\sum_{j=k+1}^{T}\sigma_j}.$$

∎

In the following lemma, we prove that the PCA reconstruction error is small due to the exponential decay of the spectrum of the Hankel covariance matrix $H$.

**Lemma** C.3. **(Uniform bound on reconstruction error)** *Under the assumptions of Lemma C.2 and for any $T \geq 10$*

$$\|\mu(\lambda) - \widetilde{\mu}(\lambda)\|_2^2 \leq C_0 T\sqrt{\log T}c^{-k/\log T},$$

*where $c = \exp(\pi^2/8)$ and $C_0 = 43$.*

*Proof.* We appeal to the following, which appears as Corollary 5.4 in Beckermann and Townsend [3].

**Lemma** C.4. *Let $H_n \in \mathbb{R}^{n \times n}$ be a positive semi-definite Hankel matrix. Then,*

$$\sigma_{j+2k} \leq 16 \Big[ \exp\Big(\frac{\pi^2}{4\log(8\lfloor n/2 \rfloor/\pi)}\Big)\Big]^{-2k+2} \sigma_j(H_n), \quad \text{for} \quad 1 \leq j+2k \leq n. \qquad (18)$$

Setting $j = 1$ in (18) with the assumption $T \geq 10$ yields

$$\sigma_{2+2k} \leq \sigma_{1+2k} \leq 16\sigma_1 \exp\Big(\frac{\pi^2}{4\log T}\Big)^{-2k+2} \leq 1168\sigma_1 \exp\Big(\frac{\pi^2}{4\log T}\Big)^{-2k}.$$

Let $c = \exp(\pi^2/8)$. It follows that

$$\sigma_j \leq 1168\sigma_1 c^{\frac{-2(j-2)}{\log T}} \leq 10512\sigma_1 c^{\frac{-2j}{\log T}}.$$

The largest singular value of Hankel matrix $H$ is bounded by

$$\sigma_1 \leq \text{tr}(H) \leq \sum_{k=1}^{T} \frac{1}{2k+1} \leq \sum_{k=1}^{T} \frac{1}{k} - 1 \leq \log T,$$

where the last inequality is due to a classic bound on the $T$-th harmonic number. We conclude from Lemma C.2 that

$$\|\mu(\lambda) - \widetilde{\mu}(\lambda)\|_2^2 \leq T\sqrt{21024\sigma_1 \sum_{j=k+1}^{T} c^{-2j/\log T}}$$

$$\leq T\sqrt{21024\log T \frac{c^{-2k/\log T}}{c^2-1}} \leq 43T\sqrt{\log T}c^{-k/\log T}.$$

$\blacksquare$

## C.2 Generalized Kolmogorov width analysis: Proof of Theorem 2

*Proof of Theorem 2.* We first prove the second claim. Let $\lambda_1, \ldots \lambda_d \in [-1,1]$ denote the eigenvalues of $G$. Let $v_i$ be the right eigenvectors of $G$ and $w_i^\top$ be the left eigenvectors of $G$. Eigendecomposition of $G^t$ implies $G^t = \sum_{i=1}^{d} v_i w_i^\top \lambda_i$. Therefore, matrix $\mu(G) = [I, G, \ldots, G^{T-1}]$ can be written as

$$\mu(G) = \sum_{i=1}^{d} v_i w_i^\top ([1, \lambda_i, \ldots, \lambda_i^{T-1}] \otimes I_d) = \sum_{i=1}^{d} v_i w_i^\top (\mu(\lambda_i) \otimes I_d),$$

where $\mu(\lambda_i) = [1, \lambda_i, \ldots, \lambda_i^{T-1}]$ is a row vector. We approximate $\mu(\lambda)$ for any $\lambda \in [-1,1]$ using principal component analysis (PCA). The covariance matrix of $\mu(\lambda)$ with respect to a uniform measure is given by

$$H = \int_{\lambda=-1}^{1} \frac{1}{2}\mu(\lambda)^\top \mu(\lambda)d\lambda \quad \Rightarrow \quad H_{ij} = \int_{-1}^{1} \frac{1}{2}\lambda^{i-1}\lambda^{j-1}d\lambda = \frac{(-1)^{i+j}+1}{2(i+j-1)}.$$

Let $\{\phi_j\}_{j=1}^{k}$ be the top $k$ eigenvectors of $H$. We approximate $\mu(\lambda)$ by $\widetilde{\mu}(\lambda) = \sum_{j=1}^{k} \langle \mu^\top(\lambda), \phi_j \rangle \phi_j^\top$:

$$\mu(G) \approx \widetilde{\mu}(G) = \sum_{i=1}^{d} v_i w_i^\top \Big(\sum_{j=1}^{k} \langle \mu^\top(\lambda), \phi_j \rangle \phi_j^\top \otimes I_d\Big)$$

$$= \sum_{j=1}^{k} \Big[\sum_{i=1}^{d} \langle \mu^\top(\lambda_i), \phi_j \rangle v_i w_i^\top\Big](\phi_j^\top \otimes I_d) = \sum_{j=1}^{k} a_j u_j.$$

Check that $a_1, \ldots, a_k \in \mathbb{R}^{d \times d}$ and $u_1, \ldots, u_k \in \mathbb{R}^{d \times dT}$. We have

$$d_k(W) = \|\mu(G) - \widetilde{\mu}(G)\|_2 = \|\sum_{i=1}^{d} v_i w_i^\top (\mu(\lambda_i) - \widetilde{\mu}(\lambda_i)) \otimes I_d\|_2$$

$$\leq \sum_{i=1}^{d} \|\mu(\lambda_i) - \widetilde{\mu}(\lambda_i)\|_2$$

$$\leq d \sup_{\lambda} \|\mu(\lambda) - \widetilde{\mu}(\lambda)\|_2.$$

The first inequality uses subadditive and submultiplicative properties of norm and that $\|v_i w_i^\top\|_2 \leq 1$, $\|I_d\|_2 = 1$. By Lemma C.3,

$$d_k(W) \leq d\sqrt{43T}(\log T)^{1/4}\Big(\exp(\pi^2/16)\Big)^{-k/\log T}.$$

Now we prove the first claim by showing that the lower bound is realized for a particular set $W$. Since the case of $d = 2$ can be embedded as a subset for general $d \geq 2$ as the left top block, it suffices to show it for $d = 2$. We further constrain the set $W$ and only consider those $G$ with representation

$$G = \begin{bmatrix} a & b \\ -b & a \end{bmatrix},$$

where $a, b \in \mathbb{R}$. The eigenvalues of this matrix are complex numbers $a - jb$ and $a + jb$, which satisfy $\rho(G) \leq 1$ if $a^2 + b^2 \leq 1$, where $\rho(G)$ is the spectral radius of $G$. The nice property of this type of matrices is that there exists an explicit expression of $G^i$ for integer $i \geq 2$. Define complex number $z = a + jb$, then for integer $i \geq 0$:

$$G^i = \begin{bmatrix} \Re(z^i) & \Im(z^i) \\ -\Im(z^i) & \Re(z^i) \end{bmatrix},$$

where $\Re(z)$ represents the real part of complex number $z$, and $\Im(z)$ represents the imaginary part of $z$.

We want to approximate $\mu(G) \in \mathbb{R}^{2 \times 2T}$ by $\sum_{i=1}^k a_i u_i$, where $a_i \in \mathbb{R}^{2 \times 2}$ and $u_i \in \mathbb{R}^{2 \times 2T}$. Let $W_1$ be the subset of row vectors realized by the first row of $\mu(G)$ for all $G \in \mathbb{R}^{2 \times 2}$ with $\rho(G) \leq 1$. We use the following property: the 2-norm of a matrix is lower bounded by the 2-norm of one of its rows. Based on this property, the 2-norm of error in approximating $\mu(G)$ is lower bounded by the 2-norm of error in approximating only one row of $\mu(G)$. Therefore, the generalized $k$-width of approximating $\mu(G)$ in 2-norm is lower bounded by the error of approximating the first row of $\mu(G)$ by a linear combination of $2k$ row vectors with dimension $2T$. In other words,

$$d_{2k}(W_1) \leq d_k(W). \tag{19}$$

To see this, denote by $u_i(1), u_i(2) \in \mathbb{R}^{2T}$ the first and second row of matrix $u_i$, respectively. The first row of $\mu(G)$ can be written as $\sum_{i=1}^k a_i(1,1)u_i(1) + a_i(1,2)u_i(2)$, a linear comibination of $2k$ row vectors, where $a_i(1,1), a_i(1,2)$ are the elements of the first row of matrix $a_i$.

To lower bound the generalized Kolmogorov width of the constrained set $W_1$, we consider a relaxed *weighted* version of the width. Precisely, let $p$ be a probability measure on the set $W_1$, then the *weighted squared deviation* of $W_1$ from $U$ under weight $p$ is defined as

$$d_{2k}^2(W_1; p) \triangleq \inf_{U \in \mathcal{U}_{2k}} \mathbb{E}_{x \sim p} \inf_{y \in U} \|x - y\|^2 \leq \inf_{U \in \mathcal{U}_{2k}} \sup_{x \in W_1} \inf_{y \in U} \|x - y\|^2 = d_{2k}^2(W_1). \tag{20}$$

We observe that $d_{2k}^2(W_1; p)$ in general can be computed using spectral methods. Indeed, for the subset $U$, the $y$ that achieves $\inf_{y \in U} \|x - y\|^2$ can be computed via a projection matrix $\hat{y} = U_{2k} U_{2k}^\top x$, where $U_{2k}$ consists of $2k$ columns of orthonormal vectors. We now have

$$\mathbb{E}_{x \sim p} \inf_{y \in Q} \|x - y\|^2 = \mathbb{E}_{x \sim p} \|x - U_{2k} U_{2k}^\top x\|^2$$

$$= \mathbb{E}_{x \sim p}[x^\top x - x^\top U_{2k} U_{2k}^\top x]$$

$$= \mathrm{tr}((I - U_{2k} U_{2k}^\top) \mathbb{E}_{x \sim p}[xx^\top]).$$

The minimizer $U_{2k}$ of $\mathrm{tr}((I - U_{2k} U_{2k}^\top) \mathbb{E}_{x \sim p}[xx^\top])$ is the same as the maximizer of $\mathrm{tr}(U_{2k} U_{2k}^\top \mathbb{E}_{x \sim p}[xx^\top])$, which is given by the first $2k$ eigenvectors of $\mathbb{E}_{x \sim p}[xx^\top]$, and the value of the weighted squared generalized $k$-width is given by the sum of all eigenvalues of $\mathbb{E}_{x \sim p}[xx^\top]$ except for the first largest $2k$ eigenvalues (Lemma C.1).

We compute the weighted squared generalized $k$-width of the constrained set $W_1$, and it would serve as a lower bound of the squared generalized $k$-width. We choose the probability measure of $(a, b)^\top \in \mathbb{R}^2$ as the uniform measure on the unit circle. We compute the matrix $\mathbb{E}_{x \sim p}[xx^\top]$,

which is $\mathbb{E}[\mu_1(G)^\top \mu_1(G)]$, where $\mu_1(G)$ is the first row of $\mu(G)$. Concretely, we write $\mu_1(G) = [\nu_0; \nu_1; \ldots; \nu_{T-1}]$ for $\nu_l \in \mathbb{R}^2$ and equal to

$$\nu_l = [\Re(z^l), \Im(z^l)],$$

where $z = a + jb$ and for all $l \in \{0, 1, \ldots, T-1\}$.

We claim that $\mathbb{E}[\nu_l \nu_m^\top] = 0$ whenever $l \neq m$. Indeed, when $l \neq m$, each of the 4 entries of matrix $\mathbb{E}[\nu_l \nu_m^\top]$ are of the form either $\Re(z^l)\Re(z^m)$, $\Im(z^l)\Im(z^m)$, or $\Re(z^l)\Im(z^m)$ for some $l \neq m$. For the complex number $z = re^{j\theta}$, we know $z^l = r^l e^{jl\theta}$ which implies that $\Re(z^l) = r^l \cos(l\theta)$ and $\Im(z^l) = r^l \sin(l\theta)$. We now compute $\mathbb{E}[r^l \cos(l\theta) r^m \sin(m\theta)]$ for $l \neq m, l \geq 0, m \geq 0$ and other cases can be computed analogously. Since we are considering a uniform distribution on the unit circle, $r \equiv 1$. We have

$$\int_{\theta \in [0, 2\pi]} \cos(k\theta) \sin(m\theta) \frac{1}{2\pi} d\theta$$
$$= \frac{1}{2\pi} \int_0^{2\pi} \frac{1}{2}(\sin((k+m)\theta) + \sin((k-m)\theta)) d\theta$$
$$= 0.$$

Hence, it suffices to only compute $\mathbb{E}[\nu_l \nu_l^\top]$

$$\mathbb{E}[\nu_l \nu_l^\top] = \frac{1}{2} \begin{bmatrix} 1 & 0 \\ 0 & 1 \end{bmatrix}. \tag{21}$$

Therefore, $\mathbb{E}[\mu_1(G)^\top \mu_1(G)] = 0.5 I_{2T}$. Using Lemma C.1 $d_{2k}^2(W_1; p = \mathcal{U})$ is equal to the sum of bottom $2T - 2k$ eigenvalues: $d_{2k}^2(W_1; p = \mathcal{U}) = (2T - 2k)/2 = T - k$. By (19) and (20)

$$d_k(W) \geq d_{2k}(W_1) \geq d_{2k}(W_1; p = \mathcal{U}) = \sqrt{T - k}.$$

$\square$

## D  Convex relaxation analysis

The approximation technique used in Theorem 2 can be applied to approximate the coefficients of the Kalman predictive model by

$$\widetilde{\mathcal{O}}_t = \sum_{j=1}^k \left[ \sum_{i=1}^d \langle \mu(\lambda_i)^\top, \phi_j \rangle C v_i w_i^\top K \right] (\phi_j^\top(t:1) \otimes I_m),$$
$$\widetilde{\mathcal{C}}_t = \sum_{j=1}^k \left[ \sum_{i=1}^d \langle \mu(\lambda_i)^\top, \phi_j \rangle C v_i w_i^\top (B - KD) \right] (\phi_j^\top(t:1) \otimes I_n),$$

where we used the fact that $[\lambda_i^{t-1}, \ldots, \lambda_i, 1]$ can be approximated by truncated eigenvectors $\{\phi_j(t:1)\}_{j=1}^k$. The relaxed model $\widetilde{m}_t \triangleq \widetilde{\mathcal{O}}_t y_{1:t-1} + \widetilde{\mathcal{C}}_t x_{1:t-1} + D x_t$ can be written in the form $\widetilde{m}_t = \widetilde{\Theta} f_t$. The feature vector $f_t$ is defined in (9) and the parameter matrix $\widetilde{\Theta}$ is obtained by concatenating the corresponding coefficient matrices as described below

$$\widetilde{\Theta} = \left[ \underbrace{\left[ \sum_{i=1}^d \langle \mu(\lambda_i)^\top, \phi_j \rangle C v_i w_i^\top K \right]_{j=1}^k}_{\substack{\in \mathbb{R}^{m \times mk} \\ \text{for output features}}} \middle| \underbrace{\left[ \sum_{i=1}^d \langle \mu(\lambda_i)^\top, \phi_j \rangle C v_i w_i^\top (B - KD) \right]_{j=1}^k}_{\substack{\in \mathbb{R}^{m \times nk} \\ \text{for input features}}} \middle| \underbrace{D}_{\substack{\in \mathbb{R}^{m \times n} \\ \text{for } x_t}} \right]_{m \times l}$$

(22)

The following theorem provides a detailed derivation of convex relaxation as well as an analysis for approximation error.

**Theorem 3. (Convex relaxation error bound)** *Denote by $m_t$, the one-step-ahead predictions made by the best linear predictor (Kalman filter) for system (4). Let $R_P = \max\{\|B\|_2, \|C\|_2, \|D\|_2\}$, $R_C = \max\{\|P\|_2, \|Q\|_2, \|R\|_2, \|K\|_2\}$, and $\|x_t\|_2 \le R_x$. Suppose that $\|A^t\|_2 \le \gamma t^{\log(\gamma)}$ for a bounded constant $\gamma \ge 1$. Let $C_0 = 43, C_1 = 520$. For any $\epsilon, \delta > 0$, if the number of filters $k$ satisfies*

$$k \ge \frac{\pi^2}{8} \log(T) \log\Big( \frac{12C_0 d^2 (1+R_P^2)^3 (2R_x^2 + R_C)(1+R_C^2)(1+\gamma)^4 (mT + \log(1/\delta))T^{3+2\log(\gamma)}}{\epsilon} \Big),$$

*then the following holds for $\widetilde{\Theta}$*

$$\mathbb{P}\Big[ \|\widetilde{\Theta} f_t - m_t\|_2^2 \ge \epsilon \Big] \le \delta. \tag{23}$$

*Proof.* Denote by $G = U\Lambda U^{-1}$ the eigendecomposition of matrix $G$, where $\Lambda = \mathrm{diag}(\lambda_1, \dots \lambda_d)$ are eigenvalues of $G$. Let $v_l$ be the columns of $U$ and $w_l^\top$ be rows of $U^{-1}$. Write

$$
\begin{aligned}
m_t &= \sum_{i=1}^{t-1} CG^{t-i-1} Ky_i + \sum_{i=1}^{t-1} CG^{t-i-1}(B-KD)x_i + Dx_t \\
&= \sum_{i=1}^{t-1} CU\Lambda^{t-i-1}U^{-1} Ky_i + \sum_{i=1}^{t-1} CU\Lambda^{t-i-1}U^{-1}(B-KD)x_i + Dx_t \\
&= \sum_{i=1}^{t-1} CU\Big[\sum_{l=1}^{d}(\lambda_l^{t-i-1})e_l \otimes e_l\Big]U^{-1} Ky_i + \sum_{i=1}^{t-1} CU\Big[\sum_{l=1}^{d}(\lambda_l^{t-i-1})e_l \otimes e_l\Big]U^{-1}(B-KD)x_i + Dx_t \\
&= \sum_{l=1}^{d} CUe_l \otimes e_l U^{-1} K \sum_{i=1}^{t-1}\lambda_l^{t-i-1}y_i + \sum_{l=1}^{d} CUe_l \otimes e_l U^{-1}(B-KD)\sum_{i=1}^{t-1}\lambda_l^{t-i-1}x_i + Dx_t \\
&= \sum_{l=1}^{d} Cv_l w_l^\top K \sum_{i=1}^{t-1}\lambda_l^{t-i-1}y_i + \sum_{l=1}^{d} Cv_l w_l^\top (B-KD)\sum_{i=1}^{t-1}\lambda_l^{t-i-1}x_i + Dx_t.
\end{aligned}
$$

Let $Y_t = [y_1, \dots, y_t] \in \mathbb{R}^{m \times t}$ and $X_t = [x_1, \dots, x_t] \in \mathbb{R}^{n \times t}$. We can write $m_t$ and $\widetilde{m}_t$ as

$$m_t = \sum_{l=1}^{d} Cv_l w_l^\top K Y_{t-1}\mu_{t-1:1}(\lambda_l) + \sum_{l=1}^{d} Cv_l w_l^\top (B-KD)X_{t-1}\mu_{t-1:1}(\lambda_l) + Dx_t,$$

$$\widetilde{m}_t = \sum_{l=1}^{d} Cv_l w_l^\top K Y_{t-1}\widetilde{\mu}_{t-1:1}(\lambda_l) + \sum_{l=1}^{d} Cv_l w_l^\top (B-KD)X_{t-1}\widetilde{\mu}_{t-1:1}(\lambda_l) + Dx_t.$$

We write $b_t = m_t - \widetilde{m}_t$ using the PCA reconstruction error $r_t = \mu_t - \widetilde{\mu}_t$

$$b_t = m_t - \widetilde{m}_t = \sum_{i=1}^{d} Cv_i w_i^\top K Y_{t-1}r_{t-1:1}(\lambda_i) + Cv_i w_i^\top (B-KD)X_{t-1}r_{t-1:1}(\lambda_i).$$

The Euclidean norm of bias is bounded by

$$
\begin{aligned}
\|b_t\|_2 &\le \Big( \sum_{i=1}^{d} \|C\|_2 \|v_i w_i^\top\|_2 \|K\|_2 \|Y_{t-1}\|_2 + \|C\|_2 \|v_i w_i^\top\|_2 (\|B\|_2 + \|K\|_2 \|D\|_2)\|X_{t-1}\|_2 \Big) \sup_\lambda \|r(\lambda)\|_2 \\
&\le \Big( dR_P R_C \|Y_{t-1}\|_2 + dR_P^2(1+R_C)\|X_{t-1}\|_2 \Big) \sup_\lambda \|r(\lambda)\|_2 \\
&\le \Big( dR_P R_C \|Y_{t-1}\|_2 + dR_P^2(1+R_C)\sqrt{t}R_x \Big) \Big( C_0 T\sqrt{\log T}c^{-k/\log T} \Big)^{1/2}.
\end{aligned}
$$

The first inequality uses simple properties such as sub-multiplicative and sub-additive properties of norm. The second inequality uses the upper bound assumptions on parameters. The third inequality is due to Lemma C.3 where $c = \exp(\pi^2/8)$ and $C_0 = 43$. The squared approximation error is given by

$$\|b_t\|_2^2 \le 2d^2(1+R_C^2)(1+R_P^2)^2\Big( \|Y_{t-1}\|_2^2 + tR_x^2 \Big) C_0 T\sqrt{\log T}c^{-k/\log T}.$$

Observe that $\|Y_{1:t}\|_2^2 \le \|Y_{1:t}\|_F^2 = \|y_{1:t}\|_2^2$. By (17), the following holds with probability greater than $1 - \delta$

$$\|b_t\|_2^2 \le 12d^2(1+R_P^2)^3(2R_x^2 + R_C)(1+R_C^2)(1+\gamma)^4(mT + \log(1/\delta))T^{3+2\log(\gamma)}c^{-k/\log T}.$$

We finish the proof by setting the number of filters $k$ such that the error is smaller than $\epsilon$, i.e.

$$k \geq \frac{\log T}{\log c} \log \Big( \frac{12 C_0 d^2 (1 + R_P^2)^3 (2R_x^2 + R_C)(1 + R_C^2)(1 + \gamma)^4 (mT + \log(1/\delta)) T^{3 + 2\log(\gamma)}}{\epsilon} \Big).$$

$\blacksquare$

## E Regret decomposition

Recall the definitions of innovation $e_t$ and model bias $b_t$

$$e_t = y_t - \mathbb{E}[y_t | y_{1:t-1}, x_{1:t}] = y_t - m_t \quad \text{and} \quad b_t = \widetilde{\Theta} f_t - m_t = \widetilde{m}_t - m_t, \quad (24)$$

where $m_t$ is the predictions made by the Kalman filter in hindsight and $\widetilde{\Theta}$ is defined in (22). Let $\hat{m}_t$ be the predictions made by the algorithm and define

$$\mathcal{L}(T) \triangleq \sum_{t=1}^{T} \|\hat{m}_t - m_t\|_2^2 \quad (25)$$

to be the squared error between the Kalman filter predictions and algorithm predictions. Regret can be written as

$$\begin{aligned}
\text{Regret}(T) &= \sum_{t=1}^{T} \|y_t - \hat{m}_t\|_2^2 - \|y_t - m_t\|_2^2 \\
&= \sum_{t=1}^{T} \|m_t + e_t - \hat{m}_t\|_2^2 - \|e_t\|_2^2 \\
&= \sum_{t=1}^{T} \|\hat{m}_t - m_t\|_2^2 - \sum_{t=1}^{T} 2e_t^\top (\hat{m}_t - m_t) \\
&= \mathcal{L}(T) - \sum_{t=1}^{T} 2e_t^\top (\hat{m}_t - m_t).
\end{aligned}$$

Recall the following notation

$$Z_t \triangleq \alpha I + \sum_{i=1}^{t} f_i f_i^\top, \quad E_t \triangleq \sum_{i=1}^{t} e_i f_i^\top, \quad B_t \triangleq \sum_{i=1}^{t} b_i f_i^\top. \quad (26)$$

The error between the predictions made by our algorithm and Kalman filter can be written as

$$\hat{m}_t - m_t = \hat{\Theta}^{(t)} f_t - \widetilde{\Theta} f_t + b_t = \Big( \sum_{i=1}^{t-1} y_i f_i^\top \Big) Z_{t-1}^{-1} f_t - \widetilde{\Theta} f_t + b_t,$$

The second equation uses the update rule of $\hat{\Theta}^{(t)}$ given in (10). Simple algebraic manipulations give

$$\begin{aligned}
&\Big( \sum_{i=1}^{t-1} y_i f_i^\top \Big) Z_{t-1}^{-1} f_t - \widetilde{\Theta} f_t + b_t \\
&= \Big( \sum_{i=1}^{t-1} [\widetilde{\Theta} f_i + b_i + e_i] f_i^\top \Big) Z_{t-1}^{-1} f_t - \widetilde{\Theta} f_t + b_t \\
&= \Big( \sum_{i=1}^{t-1} [\widetilde{\Theta} f_i f_i^\top + b_i f_i^\top + e_i f_i^\top] \Big) Z_{t-1}^{-1} f_t - \widetilde{\Theta} f_t + b_t \\
&= \Big( \sum_{i=1}^{t-1} [\widetilde{\Theta} (f_i f_i^\top + \frac{\alpha}{t-1} I - \frac{\alpha}{t-1} I) + b_i f_i^\top + e_i f_i^\top] \Big) Z_{t-1}^{-1} f_t - \widetilde{\Theta} f_t + b_t \\
&= \widetilde{\Theta} \Big( \alpha I + \sum_{i=1}^{t-1} f_i f_i^\top \Big) Z_{t-1}^{-1} f_t - \alpha \widetilde{\Theta} Z_{t-1}^{-1} f_t + \Big( \sum_{i=1}^{t-1} b_i f_i^\top \Big) Z_{t-1}^{-1} f_t + \Big( \sum_{i=1}^{t-1} e_i f_i^\top \Big) Z_{t-1}^{-1} f_t - \widetilde{\Theta} f_t + b_t \\
&= \widetilde{\Theta} Z_{t-1} Z_{t-1}^{-1} f_t - \alpha \widetilde{\Theta} Z_{t-1}^{-1} f_t + B_{t-1} Z_{t-1}^{-1} f_t + E_{t-1} Z_{t-1}^{-1} f_t - \widetilde{\Theta} f_t + b_t \\
&= E_{t-1} Z_{t-1}^{-1} f_t + B_{t-1} Z_{t-1}^{-1} f_t + b_t - \alpha \widetilde{\Theta} Z_{t-1}^{-1} f_t.
\end{aligned}$$

We apply the RMS-AM inequality to obtain an upper bound on $\mathcal{L}(T)$

$$
\begin{aligned}
\mathcal{L}(T) &= \sum_{t=1}^{T} \|\hat{m}_t - m_t\|_2^2 \\
&= \sum_{t=1}^{T} \|E_{t-1}Z_{t-1}^{-1}f_t + B_{t-1}Z_{t-1}^{-1}f_t + b_t - \alpha\widetilde{\Theta}Z_{t-1}^{-1}f_t\|_2^2 \\
&\leq \sum_{t=1}^{T} 3\|E_{t-1}Z_{t-1}^{-1}f_t\|_2^2 + 3\|B_{t-1}Z_{t-1}^{-1}f_t + b_t\|_2^2 + 3\|\alpha\widetilde{\Theta}Z_{t-1}^{-1}f_t\|_2^2.
\end{aligned}
$$

Regret can thus be decomposed to the following terms

$$
\begin{aligned}
\text{Regret}(T) \leq &\sum_{t=1}^{T} 3\|E_{t-1}Z_{t-1}^{-1}f_t\|_2^2 && \text{(least squares error)} \\
&+ \sum_{t=1}^{T} 3\|B_{t-1}Z_{t-1}^{-1}f_t + b_t\|_2^2 && \text{(improper learning bias)} \\
&+ \sum_{t=1}^{T} 3\|\alpha\widetilde{\Theta}Z_{t-1}^{-1}f_t\|_2^2 && \text{(regularization error)} \\
&- \sum_{t=1}^{T} 2e_t^{\top}(\hat{m}_t - m_t) && \text{(innovation error)}
\end{aligned}
$$

We bound each of the first three terms by extracting a $\|Z_{t-1}^{-1/2}f_t\|_2^2$, i.e. we write

$$
\|E_{t-1}Z_{t-1}^{-1}f_t\|_2^2 \leq \sup_{1 \leq t \leq T} \|E_{t-1}Z_{t-1}^{-1/2}\|_2^2 \sum_{t=1}^{T} \|Z_{t-1}^{-1/2}f_t\|_2^2,
$$

$$
\|B_{t-1}Z_{t-1}^{-1}f_t + b_t\|_2^2 \leq \sup_{1 \leq t \leq T} \|B_{t-1}Z_{t-1}^{-1/2}\|_2^2 \sum_{t=1}^{T} \|Z_{t-1}^{-1/2}f_t\|_2^2 + \sum_{t=1}^{T} \|b_t\|_2^2,
$$

$$
\|\alpha\widetilde{\Theta}Z_{t-1}^{-1}f_t\|_2^2 \leq \sup_{1 \leq t \leq T} \|\alpha\widetilde{\Theta}Z_{t-1}^{-1/2}\|_2^2 \sum_{t=1}^{T} \|Z_{t-1}^{-1/2}f_t\|_2^2.
$$

In subsequent sections, we compute a high probability upper bound on $\sum_{t=1}^{T} \|Z_{t-1}^{-1/2}f_t\|_2^2$ as well as the specific terms in the above decomposition that affect least squares error, improper learning bias, regularization error, and innovation error, proving that regret is bounded by $\text{polylog}(T)$.

## F   Regret analysis

**PAC bound parameters notation.**   We define $M = (R_\Theta, m, \gamma, \kappa, \beta, \gamma, \delta)$ to be a shorthand for the PAC bound parameters (defined in Theorem 1). Given a function $f : \mathbb{N} \to R$, we write $x \lesssim_M f(T), x \asymp_M f(T)$ to specify the dependency only on the horizon $T$.

### F.1   High probability bound on $\det(Z_t)$

We start by deriving an upper bound on $\log(\det(Z_t))$ as this quantity appears multiple times when analyzing regret. The following lemma provides a high probability bound on $\det(Z_t)$ for features defined in (9).

**Lemma** F.1. (**High probability upper bounds on $\det(\mathbf{Z_t})$**) *Assume as in Lemma B.2 and let $Z_t = \alpha I + \sum_{i=1}^{t} f_t f_t^{\top}$. Then, for any $\delta \geq 0$*

$$
\mathbb{P}\left(\log(\det(Z_t)) \geq l \log\left[\alpha^2 + 8k(R_P^2 + 1)(R_x^2 + R_C)(1 + \gamma)^4(mt + \log\left(\frac{1}{\delta}\right))t^{3+2\log(\gamma)}\right]\right) \leq \delta.
$$

*Proof.* Let $l$ be the feature vector dimension. We have

$$Z_t = \alpha I + \sum_{i=1}^{t} f_t f_t^\top \preceq \alpha I + \sum_{i=1}^{t}(f_i^\top f_i)I \quad \Rightarrow \quad \det(Z_t) \leq \left(\alpha^2 + \sum_{i=1}^{t} \|f_i\|_2^2\right)^l.$$

Recall the definition $\Psi_t = [\psi_t, \ldots, \psi_1]$ from Algorithm 1 and the compact representation for input features $\widetilde{x}_t = (\Psi_t \otimes I_n)x_{1:t}$ and output features $\widetilde{y}_t = (\Psi_t \otimes I_n)y_{1:t}$. Observe that $\|\Psi_t\|_2 \leq 1$ since $\Psi_t$ is a block of eigenvector matrix of hankel matrix $H$. Thus the feature norm is bounded by

$$\begin{aligned}
\|f_t\|_2^2 &= \|\widetilde{y}_{t-1}\|_2^2 + \|\widetilde{x}_{t-1}\|_2^2 + \|x_t\|_2^2 \\
&\leq k\|y_{1:t-1}\|_2^2 + k\|x_{1:t-1}\|_2^2 + R_x^2 \\
&\leq k\|y_{1:t}\|_2^2 + 2ktR_x^2.
\end{aligned}$$

From Lemma B.2, with probability at least $1 - \delta$

$$\begin{aligned}
\|f_t\|_2^2 &\leq 6k(R_P^2 + 1)(R_x^2 + R_C)(1 + \gamma)^4(mt + \log\left(\tfrac{1}{\delta}\right))t^{2+2\log(\gamma)} + 2ktR_x^2 \\
&\leq 8k(R_P^2 + 1)(R_x^2 + R_C)(1 + \gamma)^4(mt + \log\left(\tfrac{1}{\delta}\right))t^{2+2\log(\gamma)}.
\end{aligned}$$

The above bound is increasing in $t$, therefore

$$\mathbb{P}\left(\det(Z_t) \geq \left[\alpha^2 + 8k(R_P^2 + 1)(R_x^2 + R_C)(1 + \gamma)^4(mt + \log\left(\tfrac{1}{\delta}\right))t^{3+2\log(\gamma)}\right]^l\right) \leq \delta.$$

$\blacksquare$

Given the PAC bound parameters $M$, if $k \asymp_M \text{polylog}(T)$ (and hence $l = (m+n)k + n \asymp_M \text{polylog}(T)$), then the above lemma states that $\log(\det(Z_t)) \lesssim_M \text{polylog}(T)$.

## F.2 Self-normalizing vector martingales

We now prove a key result on vector self-normalizing martingales that is used multiple times throughout our regret analysis. The result is inspired by Theorem 1 of Abbasi-Yadkori et al. [1], which provides a bound for self-normalizing martingales with scalar sub-Gaussian noise, and extend it to vector-valued sub-Gaussian noise with arbitrary covariance.

**Theorem** F.1. **(Bound on self-normalized vector martingale)** *Let $\{\mathcal{F}_t\}_{t=0}^\infty$ be a filtration. Let $e_t \in \mathbb{R}^m$ be $\mathcal{F}_t$ measurable and $e_t|\mathcal{F}_{t-1}$ to be conditionally $R_V$-sub-Gaussian. In other words, for all $t \geq 0$ and $\omega \in \mathbb{R}^m$*

$$\mathbb{E}[\exp(\omega^\top e_t) \mid \mathcal{F}_{t-1}] \leq \exp(R_V^2 \|\omega\|_2^2/2).$$

*Let $f_t \in \mathbb{R}^l$ be an $\mathcal{F}_{t-1}$-measurable stochastic process. Assume that $Z$ is an $l \times l$ positive definite matrix. For any $t \geq 0$, define*

$$Z_t = Z_0 + \sum_{i=1}^{t} f_t f_t^\top \quad \text{and} \quad E_t = \sum_{i=1}^{t} e_i f_i^\top.$$

*Then, for any $\delta > 0$ and for all $t \geq 0$*

$$\mathbb{P}\left[\|E_t Z_t^{-1/2}\|_2 \leq 8R_V^2 m + 4R_V^2 \log\left(\frac{\det(Z_t)^{1/2}\det(Z_0)^{-1/2}}{\delta}\right)\right] \geq 1 - \delta.$$

*Proof.* We use an $\epsilon$-net argument. First, we establish control over $\|\omega^\top E_t Z_t^{-1/2}\|_2$ for all vectors $\omega$ in unit sphere $\mathcal{S}^{m-1}$. We will discretize the sphere using a net and finish by taking a union bound over all $\omega$ in the net.

Let $\mathcal{N}$ be an $\epsilon$-net of unit sphere $\mathcal{S}^{m-1}$ and set $\epsilon = 1/2$. Corollary 4.2.13 in [13] states that the covering number for unit sphere $\mathcal{S}^{m-1}$ is given by

$$|\mathcal{N}| \leq \left(\frac{2}{\epsilon} + 1\right)^m = 5^m.$$

$\omega^\top e_i$ is $R_V$-sub-Gaussian for any $\omega \in \mathcal{N}$. Therefore, for any $\omega \in \mathcal{N}$ and any $u \geq 0$, Theorem 1 in [1] yields

$$\mathbb{P}\left[\|\omega^\top E_t Z_t^{-1/2}\|_2 \geq u\right] \leq \det(Z_t)^{1/2} \det(Z_0)^{-1/2} \exp\left(-\frac{u}{2R_V^2}\right).$$

Using Lemma 4.4.1 in [13], we have

$$\|E_t Z_t^{-1/2}\|_2 \leq 2 \sup_{\omega \in \mathcal{N}} \|\omega^\top E_t Z_t^{-1/2}\|_2.$$

Taking a union bound over $\mathcal{N}$, we conclude that

$$
\begin{aligned}
\mathbb{P}\left[\|E_t Z_t^{-1/2}\|_2 \geq u\right] &\leq \mathbb{P}\left[\sup_{\omega \in \mathcal{N}} \|\omega^\top E_t Z_t^{-1/2}\|_2 \geq \frac{u}{2}\right] \\
&\leq \sum_{\omega \in \mathcal{N}} \mathbb{P}\left[\|\omega^\top E_t Z_t^{-1/2}\|_2 \geq \frac{u}{2}\right] \\
&\leq \det(Z_t)^{1/2} \det(Z_0)^{-1/2} \exp\left(2m - \frac{u}{4R_V^2}\right).
\end{aligned}
$$

∎

The above theorem combined with the result of Lemma F.1 immediately implies that for $k \asymp_M$ polylog$(T)$, we have $\|E_t Z_t^{-1/2}\|_2 \lesssim_M$ polylog$(T)$ and $\|B_t Z_t^{-1/2}\|_2 \lesssim_M$ polylog$(T)$ with high probability.

### F.3 High probability bound on $\|Z_{t-1}^{-1/2} f_t\|_2^2$

In this section, we show that $\sum_{t=1}^T \|Z_{t-1}^{-1/2} f_t\|_2^2 \lesssim_M$ polylog$(T)$. The proof steps are summarized below.

Step 1. We show a high probability Löwner upper bound on $f_t f_t^\top$ in terms of $\alpha_0 I + \mathbb{E}[f_t f_t^\top]$.

Step 2. We state the *block-martingale small-ball condition* and show that the process $\{f_t\}$ satisfies this condition. We prove a high probability lower bound on $Z_t$ in terms of the conditional covariance cov$(f_{s+i} \mid \mathcal{F}_i)$ for large enough $s$.

Step 3. We define a *filter quadratic function condition* and prove that under this condition, there exists $c_T \asymp_M$ polylog$(T)$ such that $Z_t - \frac{1}{c_T} f_{t+1} f_{t+1}^\top \succeq 0$. By Schur complement lemma, this is equivalent to $\|Z_t^{-1/2} f_{t+1}\|_2 \leq c_T \asymp_M$ polylog$(T)$.

**Step 1.** The following lemma establishes a high probability upper bound on $f_t f_t^\top$ based on the covariance of feature vector $f_t$.

**Lemma F.2. (High probability upper bound on $\mathbf{f_t f_t^\top}$)** *Let $f_t$ be a zero-mean Gaussian random vector in $\mathbb{R}^l$ and let $\Sigma_t = \alpha_0 I + \mathbb{E}[f_t f_t^\top]$ for a real $\alpha_0 > 0$. Then, for any $\delta > 0$ and $\alpha_0 > 0$*

$$\mathbb{P}\left(f_t f_t^\top \preceq [2l + 4\log(1/\delta)]\Sigma_t\right) \geq 1 - \delta,$$

*and if $\Sigma_t$ is invertible, the results holds for $\alpha_0 = 0$.*

*Proof.* Consider the random vector $\Sigma_t^{-1/2} f_t$. Jensen's inequality gives

$$\mathbb{E}\|\Sigma_t^{-1/2} f_t\|_2 \leq \sqrt{\mathbb{E}[f_t^\top \Sigma_t^{-1} f_t]} = \sqrt{\mathrm{tr}(\Sigma_t^{-1} \mathbb{E}[f_t f_t^\top])} \leq \sqrt{l}.$$

By standard bounds on tails of sub-gaussian random variables (for example, see Exercise 6.3.5 in [13]), for any $\delta > 0$

$$\mathbb{P}\left(\|\Sigma_t^{-1/2} f_t\|_2 > \sqrt{l} + \sqrt{2\log \frac{1}{\delta}}\right) \leq \delta$$

Let $c = 2l + 4\log\frac{1}{\delta}$. Then, the above bound implies

$$\mathbb{P}(f_t^\top \Sigma_t^{-1} f_t \leq c) \geq 1 - \delta.$$

Using Schur complement method, $c - f_t^\top \Sigma_t^{-1} f_t \geq 0$ if and only if the following matrix is positive semi-definite

$$\begin{bmatrix} \Sigma_t & f_t \\ f_t^\top & c \end{bmatrix} \succeq 0.$$

Using the other Schur complement, this is only true if and only if $\Sigma_t - \frac{1}{c} f_t f_t^\top \succeq 0$, which concludes the proof. ∎

**Step 2.** To capture the excitation behavior of features, we use the martingale small-ball condition [8, 10].

**Definition** 1. **(Martingale small-ball)** *Let $\{f_t\}_{t\geq 1}$ be an $\mathcal{F}_t$-adapted random processes taking values in $\mathbb{R}^l$. We say that $\{f_t\}_{t\geq 1}$ satisfies the $(s, \Gamma_{sb}, p)$-block martingale small-ball (BMSB) condition for $\Gamma_{sb} \succ 0$ if for any $t \geq 1$ and for any fixed $\omega$ in unit sphere $\mathcal{S}^{l-1}$*

$$\frac{1}{s}\sum_{i=1}^{s}\mathbb{P}(|w^\top f_{t+i}| \geq \sqrt{w^\top \Gamma_{sb} w} \mid \mathcal{F}_t) \geq p.$$

To show the process $\{f_t\}_{t\geq 1}$ satisfy a BMSB condition, we first show that the conditional covariance of features is increasing in the positive semi-definite cone.

**Lemma** F.3. **(Monotonicity of conditional covariance of features)** *Let $\phi_1, \ldots, \phi_k$ for $k \leq T$ be a set of $T$-dimensional orthogonal vectors and let $\psi_i = [\phi_1(i), \ldots, \phi_k(i)]^\top$ be a $k$-dimensional vector. Consider system (4) and define the following for all $t \geq 2$*

$$f_t = \psi_1 \otimes y_{t-1} + \cdots + \psi_{t-1} \otimes y_1. \tag{27}$$

*Let $\mathcal{F}_t = \sigma\{\eta_0, \ldots, \eta_{t-1}, \zeta_1, \ldots, \zeta_t\}$. Then, $\mathrm{cov}(f_{t+i}|\mathcal{F}_t)$ is independent of $t$ and increases with $i$ in the positive semi-definite cone.*

*Proof.* Expanding $y_i$ in definition of $f_t$ in (27) based on system (4), we have

$$
\begin{aligned}
f_{t+i} - \mathbb{E}[f_{t+i} \mid \mathcal{F}_t] =\ & (\psi_1 \otimes C)\eta_{t+i-2} \\
& + (\psi_2 \otimes C + \psi_1 \otimes CA)\eta_{t+i-3} \\
& + \ldots \\
& + (\psi_{i-1} \otimes C + \cdots + \psi_1 \otimes CA^{i-2})\eta_t \\
& + \psi_1 \otimes \zeta_{t+i-1} + \cdots + \psi_{i-1} \otimes \zeta_{t+1}
\end{aligned}
$$

Recall that $\mathbb{E}[\eta_t \eta_t^\top] = Q, \mathbb{E}[\zeta_t \zeta_t^\top] = R$ and that the process noise and the observation noise are i.i.d. Therefore,

$$
\begin{aligned}
\mathrm{cov}(f_{t+i}|\mathcal{F}_t) =\ & (\psi_1 \otimes C)Q(\psi_1 \otimes C)^\top \\
& + (\psi_2 \otimes C + \psi_1 \otimes CA)Q(\psi_2 \otimes C + \psi_1 \otimes CA)^\top \\
& + \ldots \\
& + (\psi_{i-1} \otimes C + \cdots + \psi_1 \otimes CA^{i-2})Q(\psi_{i-1} \otimes C + \cdots + \psi_1 \otimes CA^{i-2})^\top \\
& + \psi_1 \otimes R\psi_1^\top \otimes I_m + \cdots + \psi_{i-1} \otimes R\psi_{i-1}^\top \otimes I_m.
\end{aligned} \tag{28}
$$

Observe that the conditional covariance is independent of $t$. Furthermore, all terms in the above sum are positive semi-definite; increasing $i$ only adds two additional positive semi-definite terms. It follows that

$$\mathrm{cov}(f_{t+i+1}|\mathcal{F}_t) \succeq \mathrm{cov}(f_{t+i}|\mathcal{F}_t).$$

∎

Equipped with the result of the above lemma, we now show that $\{f_t\}_{t\geq 1}$ satisfy a BMSB condition.

**Lemma** F.4. **(BMSB condition)** *Consider the process $\{f_t\}_{t \geq 1}$ defined in Lemma F.3 and let $\Gamma_i = cov(f_{t+i}|\mathcal{F}_t)$. For any $1 \leq s \leq T$, the process $\{f_t\}_{t \geq 1}$ satisfies the $(s, \Gamma_{s/2}, 3/20)$-BMSB condition.*

*Proof.* Note that $\omega^\top f_{t+i} \mid \mathcal{F}_t$ has a Gaussian distribution with variance $\sqrt{\omega^\top \Gamma_i \omega}$. By an application of Paley-Zygmund inequality, one has

$$\mathbb{P}(|w^\top f_{t+i}| \geq \sqrt{w^\top \Gamma_i w} \mid \mathcal{F}_t) \geq \mathbb{P}(|w^\top f_{t+i} - \mathbb{E}[w^\top f_{t+i} \mid \mathcal{F}_t]| \geq \sqrt{w^\top \Gamma_i w} \mid \mathcal{F}_t) \geq \frac{3}{10}$$

Let $1 \leq s' \leq s$. By Lemma F.3, $\Gamma_i$ is increasing in $i$. Therefore,

$$\frac{1}{s} \sum_{i=1}^{s} \mathbb{P}(|w^\top f_{t+i}| \geq \sqrt{w^\top \Gamma_{s'} w}|\mathcal{F}_t) \geq \frac{1}{s} \sum_{i=s'}^{s} \mathbb{P}(|w^\top f_{t+i}| \geq \sqrt{w^\top \Gamma_{s'} w}|\mathcal{F}_t)$$

$$\geq \frac{1}{s} \sum_{i=s'}^{s} \mathbb{P}(|w^\top f_{t+i}| \geq \sqrt{w^\top \Gamma_i w}|\mathcal{F}_t) \quad (\Gamma_i \text{ increasing})$$

$$\geq \frac{3}{10} \frac{s - s' + 1}{s}. \quad \text{(Paley-Zygmund)}$$

Choosing $s' = s/2$ shows that $f_t$ satisfies $(s, \Gamma_{s/2}, 3/20)$ small-ball condition. ∎

The small-ball condition can be used to establish high probability lower bound on $\sigma_{\min}(Z_t)$, as shown by the following lemma.

**Lemma** F.5. **(Lower bound on $Z_t$)** *Consider the process $\{f_t\}_{t \geq 1}$ defined in Lemma F.3 and let $Z_t = \alpha I + \sum_{i=1}^{t} f_i f_i^\top$ for regularization parameter $\alpha > 0$. For $\delta, \alpha_0 > 0$ let*

$$\Gamma_i = \mathrm{cov}(f_{t+i}|\mathcal{F}_i), \quad \Gamma_{\max} = t[2l + 4\log(2/\delta)][\alpha_0 I + \Gamma_t].$$

*For any $\delta > 0$ if $s$ satisfies the following*

$$s \leq \frac{tp^2/10}{\log \det(\Gamma_{\max}) - l\log(\alpha) - \log(2/\delta)},$$

*then*

$$\mathbb{P}\left(Z_t \succeq \frac{\alpha}{2} I + \frac{s\lfloor t/s \rfloor p^2 \Gamma_{s/2}}{16}\right) \geq 1 - \delta.$$

*Proof.* According to Lemma F.4, $\{f_t\}_{t \geq 1}$ satisfies the $(s, \Gamma_{s/2}, p = 3/20)$-BMSB condition. The following lemma from [10] gives tail probabilities for real-valued processes that satisfy a small-ball condition. Note that our notation for small ball condition in real-valued processes slightly differs from [10] which results in a slight difference in the statement of the lemma below.

**Lemma** F.6. **(Tail bounds for small-ball processes)** *If a real-valued process $\{z_t\}_{t \geq 1}$ satisfies the $(s, \sigma, p)$-BMSB condition, then*

$$\mathbb{P}(\sum_{i=1}^{t} z_i^2 \leq \frac{p^2 \sigma}{8} s\lfloor t/s \rfloor) \leq \exp\left(-\frac{\lfloor t/s \rfloor p^2}{8}\right).$$

For a fixed $\omega \in \mathcal{S}^{l-1}$, the process $\{\omega^\top f_t\}_{t \geq 1}$ satisfies $(s, \omega^\top \Gamma_{s/2} \omega, p)$. Using the above lemma, we have

$$\mathbb{P}\left(\omega^\top \left(\sum_{i=1}^{t} f_i f_i^\top\right)\omega \leq \frac{p^2 \omega^\top \Gamma_{s/2} \omega}{8} s\lfloor t/s \rfloor\right) \leq \exp\left(-\frac{\lfloor t/s \rfloor p^2}{8}\right).$$

For large enough $t$, we can convert this high probability bound to obtain a uniform Löwner lower bound on $Z_t$ by a discretization argument.

Given a regularization parameter $\alpha > 0$, define

$$\Gamma_{\min} = \alpha I + \frac{s\lfloor t/s \rfloor p^2 \Gamma_{s/2}}{8}$$

Define the following events

$$\mathcal{E}_1 = \left\{ Z_t \succeq \frac{\Gamma_{\min}}{2} \right\} \quad \text{and} \quad \mathcal{E}_2 = \left\{ Z_t \preceq \Gamma_{\max} \right\}.$$

We have $\mathbb{P}(\mathcal{E}_1^c) \leq \mathbb{P}(\mathcal{E}_1^c \cap \mathcal{E}_2) + \mathbb{P}(\mathcal{E}_2^c)$, where $\mathbb{P}(\mathcal{E}_2^c)$ is bounded by $\delta/2$ according to Lemma F.2. Let $\mathcal{S}_{\Gamma_{\mathrm{sb}}} = \{\omega : \omega^\top \Gamma_{\mathrm{sb}} \omega = 1\}$ and let $\mathcal{T}$ be a $1/4$-net of $\mathcal{S}_{\Gamma_{\mathrm{sb}}}$ in the norm $\|\Gamma_{\max}^{1/2}(.)\|_2$. By Lemma 4.1 and Lemma D.1 in [10], we can write

$$
\begin{aligned}
\mathbb{P}(\mathcal{E}_1^c \cap \mathcal{E}_2) &= \mathbb{P}\left( \left\{ Z_t \not\succeq \frac{\Gamma_{\min}}{2} \right\} \cap \left\{ Z_t \preceq \Gamma_{\max} \right\} \right) \\
&\leq \mathbb{P}\left( \left\{ \exists \omega \in \mathcal{T} : \|Z_t \omega\|^2 < \omega^\top \Gamma_{\min} \omega \right\} \cap \left\{ Z_t \preceq \Gamma_{\max} \right\} \right) \\
&\leq \exp\left( -\frac{\lfloor t/s \rfloor p^2}{8} + \log \det(\Gamma_{\max} \Gamma_{\min}^{-1}) \right) \\
&\leq \exp\left( -\frac{tp^2}{10s} + \log \frac{\det(\Gamma_{\max})}{\alpha^l} \right)
\end{aligned}
$$

Setting $s$ such that the above probability is bounded by $\delta/2$

$$s \leq \frac{tp^2/10}{\log \det(\Gamma_{\max}) - l \log(\alpha) + \log(2/\delta)},$$

we conclude that $\mathbb{P}(\mathcal{E}_1^c) \leq \delta/2 + \delta/2 = \delta$. ∎

**Step 3.** So far we have computed a lower bound on $Z_t$ and an upper bound on $f_t f_t^\top$ and our goal is to show that there exists $c_T \asymp_M \mathrm{polylog}(T)$ such that $Z_{t-1} - \frac{1}{c_T} f_t f_t^\top \succeq 0$. This inequality, however, does not hold for any set of orthonormal filters $\phi_1, \ldots, \phi_k$. We identify an assumption connecting filters with transition matrix $A$ that ensures $Z_{t-1} - \frac{1}{c_T} f_t f_t^\top \succeq 0$. This assumption is based on a *filter quadratic function*, which we restate below.

**Definition 2. (Filter quadratic function)** Let $\phi_1, \ldots, \phi_k$ for $k \leq T$ be a set of $T$-dimensional vectors, let $\psi_i = [\phi_1(i), \ldots, \phi_k(i)]^\top$ be a $k$-dimensional vector, and let $\psi_i^{(d)} = \psi_i \otimes I_d$, for any $d \geq 1$. For any matrix $A \in \mathbb{R}^{d \times d}$, the following matrix is called the *filter quadratic function* of $\psi$ with respect to $A$

$$
\begin{aligned}
\Omega_t(A; \psi) &= (\psi_1^{(d)})(\psi_1^{(d)})^\top + (\psi_2^{(d)} + \psi_1^{(d)} A)(\psi_2^{(d)} + \psi_1^{(d)} A)^\top + \ldots \\
&\quad + (\psi_{t-1}^{(d)} + \cdots + \psi_1^{(d)} A^{t-2})(\psi_{t-1}^{(d)} + \cdots + \psi_1^{(d)} A^{t-2})^\top.
\end{aligned}
$$

In the following lemma, we show that a condition on filter quadratic function implies $t\Gamma_{s/2} - \Gamma_{t+1}/c_0 \succeq 0$ for a constant $c_0$.

**Lemma F.7. (Filter quadratic condition)** *Assume as in Lemma F.3 and let $\kappa$ be the maximum condition number of $Q$ and $R$. For any $A$, if there exists $t_0 \geq 1$ for which there exists $s$ such that*

$$t\Omega_{s/2}(A; \psi) - \Omega_{t+1}(A; \psi) \succeq 0, \qquad \forall t \geq t_0,$$

*then $t\Gamma_{s/2} - \Gamma_{t+1}/c_0 \succeq 0$, where $c_0 \geq \kappa$.*

*Proof.* Let $\psi_i^{(m)} = \psi_i \otimes I_m$. Recall the expression of the conditional covariance of $f_t$ given in (28):

$$
\begin{aligned}
\Gamma_t &= (\psi_1^{(m)} C) Q (\psi_1^{(m)} C)^\top \\
&\quad + (\psi_2^{(m)} C + \psi_1^{(m)} CA) Q (\psi_2^{(m)} C + \psi_1^{(m)} CA)^\top \\
&\quad + \ldots \\
&\quad + (\psi_{t-1}^{(m)} C + \cdots + \psi_1^{(m)} CA^{t-2}) Q (\psi_{t-1}^{(m)} C + \cdots + \psi_1^{(m)} CA^{t-2})^\top \\
&\quad + \psi_1^{(m)} R (\psi_1^{(m)})^\top + \cdots + \psi_{t-1}^{(m)} R (\psi_{t-1}^{(m)})^\top
\end{aligned}
$$

Define the following terms

$$\Gamma_t^{(Q)} \triangleq (\psi_1^{(m)} C) Q (\psi_1^{(m)} C)^\top + \cdots + (\psi_{t-1}^{(m)} C + \cdots + \psi_1^{(m)} CA^{t-2}) Q (\psi_{t-1}^{(m)} C + \cdots + \psi_1^{(m)} CA^{t-2})^\top,$$

$$\Gamma_t^{(R)} \triangleq \psi_1^{(m)} R (\psi_1^{(m)})^\top + \cdots + \psi_{t-1}^{(m)} R (\psi_{t-1}^{(m)})^\top,$$

where $\Gamma_t = \Gamma_t^{(Q)} + \Gamma_t^{(R)}$. In order to show $t\Gamma_{s/2} - \Gamma_{t+1}/c_0 \succeq 0$, it is sufficient to show

$$t\Gamma_{s/2}^{(Q)} - \frac{1}{c_0}\Gamma_{t+1}^{(Q)} \succeq 0 \quad \text{and} \quad t\Gamma_{s/2}^{(R)} - \frac{1}{c_0}\Gamma_{t+1}^{(R)} \succeq 0.$$

Let $R_C = \max\{\|R\|_2, \|Q\|_2\}$ and $\sigma_r = \min\{\sigma_{\min}(Q), \sigma_{\min}(R)\}$. For $t\Gamma_{s/2}^{(R)} - \frac{1}{c_0}\Gamma_{t+1}^{(R)}$, we have

$$\sigma_r[\psi_1^{(m)}(\psi_1^{(m)})^\top + \cdots + \psi_t^{(m)}(\psi_t^{(m)})^\top]$$
$$\preceq \psi_1^{(m)} R(\psi_1^{(m)})^\top + \cdots + \psi_t^{(m)} R(\psi_t^{(m)})^\top$$
$$\preceq R_C[\psi_1^{(m)}(\psi_1^{(m)})^\top + \cdots + \psi_t^{(m)}(\psi_t^{(m)})^\top].$$

Setting $c_0 = R_C/\sigma_r$, gives

$$t\Gamma_{s/2}^{(R)} - \frac{1}{c_0}\Gamma_{t+1}^{(R)}$$
$$\succeq \sigma_r t[\psi_1^{(m)}(\psi_1^{(m)})^\top + \cdots + \psi_{s/2-1}^{(m)}(\psi_{s/2-1}^{(m)})^\top] - \sigma_r[\psi_1^{(m)}(\psi_1^{(m)})^\top + \cdots + \psi_t^{(m)}(\psi_t^{(m)})^\top] \succeq 0.$$

The last matrix is positive semi-definite based on assumption (29) when $A = 0$. For $t\Gamma_{s/2}^{(Q)} - \frac{1}{c_0}\Gamma_{t+1}^{(Q)}$, write

$$\psi_i^{(m)} C = \begin{bmatrix} \phi_i^1 C \\ \phi_i^2 C \\ \vdots \\ \phi_i^k C \end{bmatrix}_{km \times d} = \begin{bmatrix} C & 0 & \dots & 0 \\ 0 & C & \dots & 0 \\ \vdots & & & \\ 0 & 0 & \dots & C \end{bmatrix}_{km \times kd} \begin{bmatrix} \phi_i^1 I_d \\ \phi_i^2 I_d \\ \vdots \\ \phi_i^k I_d \end{bmatrix}_{kd \times d} = \mathbf{C}\psi_i^{(d)}.$$

We have
$$\Gamma_{t+1}^{(Q)} = \mathbf{C}\left[\psi_1^{(d)} Q(\psi_1^{(d)})^\top + \cdots + (\psi_t^{(d)} + \cdots + \psi_1^{(d)} A^{t-1}) Q(\psi_t^{(d)} + \cdots + \psi_1^{(d)} A^{t-1})^\top\right]\mathbf{C}^\top$$

By a similar argument and given assumption (29), we have $t\Gamma_{s/2}^{(Q)} - \frac{1}{c_0}\Gamma_{t+1}^{(Q)} \succeq 0$. ∎

**Remark 3.** When $A$ is symmetric ($A = UDU^\top$), the positive semi-definite condition filter quadratic function can be further simplified to $t\Omega_{s/2}(D; \psi) - \Omega_{t+1}(D; \psi) \succeq 0$ for all diagonal matrices $D$ with $|D_{ii}| \leq 1$.

In the following lemma, we show a high probability upper bound on $\|Z_t^{-1/2} f_{t+1}\|_2$.

**Lemma** F.8. ($\|\mathbf{Z_t}^{-1/2}\mathbf{f_{t+1}}\|_2$ **upper bound**) *Assume as in Lemma F.3 and let $\kappa$ be the maximum condition number of $Q$ and $R$. Define the following for all $t \geq 1$, regularization parameter $\alpha > 0$, $p = 3/20$, and fix $0 < \alpha_0 \leq 200\alpha$ and $\delta > 0$*

$$Z_t = \alpha I + \sum_{i=1}^{t} f_i f_i^\top, \quad \Gamma_{\max} = t[2km + 4\log(4/\delta)][\alpha_0 I + \Gamma_t], \quad \Gamma_{\min} = \alpha I + \frac{s\lfloor t/s \rfloor p^2 \Gamma_{s/2}}{8}$$

*For any A, suppose that there exists $t_0 \geq 1$ for which there exists $s$ such that*

$$s \leq \frac{tp^2/10}{\log\det(\Gamma_{\max}) - l\log(\alpha) + \log(4/\delta)}, \quad t\Omega_{s/2}(A; \psi) - \Omega_{t+1}(A; \psi) \succeq 0. \quad (29)$$

*Then, for all $t \geq t_0$ with probability at least $1 - \delta$*

$$\|Z_{t-1}^{-1/2} f_t\|_2^2 \leq 10\kappa(2mk + 4\log(2/\delta))/p^2.$$

*Proof.* Let $c_T = 10\kappa(2mk + 4\log(2/\delta))/p^2$. With probability at least $1 - \delta$, we lower bound $\sum_{i=1}^{t} f_i f_i^\top$ by Lemma F.5 and upper bound $\frac{1}{c_T} f_{t+1} f_{t+1}^\top$ by Lemma F.2

$$Z_t - \frac{1}{c_T} f_{t+1} f_{t+1}^\top = \alpha I + \sum_{i=1}^{t} f_i f_i^\top - \frac{1}{c} f_{t+1} f_{t+1}^\top$$
$$\succeq \frac{\alpha}{2} I + \frac{p^2}{10} t\Gamma_{s/2} - \frac{p^2}{10}\alpha_0 I - \frac{p^2}{10}\frac{1}{c_0}\Gamma_{t+1}$$
$$\overset{(1)}{\succeq} + \frac{p^2}{10} t\Gamma_{s/2} - \frac{p^2}{10}\frac{1}{c_0}\Gamma_{t+1}$$
$$\overset{(2)}{\succeq} 0$$

where inequality (1) is due to the assumption $\alpha_0 \leq 200\alpha$ and (2) uses the result of Lemma F.7.

Using Schur complement lemma, $Z_t - \frac{1}{c_T} f_{t+1} f_{t+1}^\top$ is positive semi-definite if and only if the following matrix is positive semi-definite

$$\begin{bmatrix} Z_t & f_{t+1} \\ f_{t+1}^\top & c_T. \end{bmatrix}.$$

Using the other Schur complement, this is true if and only if $c_T - f_{t+1}^\top Z_t^{-1} f_t \geq 0$. Equivalently,

$$Z_t - \frac{1}{c_T} f_{t+1} f_{t+1}^\top \succeq 0 \quad \Leftrightarrow \quad \|Z_t^{-1/2} f_{t+1}\|_2 \leq c_T,$$

which concludes the proof. ∎

The above lemma states that if $k \asymp_M \text{polylog}(T)$ then $\|Z_{t-1}^{-1/2} f_t\|_2^2 \lesssim_M \text{polylog}(T)$ with high probability.

## F.4 Proof of Lemma 1

We now prove that $\|Z_{t-1}^{-1/2} f_t\|_2^2 \lesssim_M \text{polylog}(T)$ implies $\sum_{t=1}^T \|Z_{t-1}^{-1/2} f_t\|_2^2 \lesssim_M \text{polylog}(T)$. We first present a lemma inspired by Lemma 2 in [7].

**Lemma F.9. (Upper bound on $\sum_{i=1}^t \|Z_i^{-1/2} f_i\|_2^2$)** *Let $f_1, \ldots, f_t$ be $l$-dimensional vectors and $Z_0$ an $l \times l$ positive definite matrix. Define $Z_t = Z_0 + \sum_{i=1}^t f_i f_i^\top$. Then,*

$$\sum_{i=1}^t f_i^\top Z_i^{-1} f_i \leq \log\left(\frac{\det(Z_t)}{\det(Z_0)}\right).$$

*Proof.* First, note that $Z_t$ is positive definite and has a positive determinant for all $t \geq 1$. Using matrix determinant lemma, we have

$$\det(Z_{t-1}) = \det(Z_t - f_t f_t^\top) = \det(Z_t)(1 - f_t^\top Z_t^{-1} f_t) \Rightarrow f_t^\top Z_t^{-1} f_t = \frac{\det(Z_t) - \det(Z_{t-1})}{\det(Z_t)}$$

Since $Z_i \succeq Z_{i-1}$, we have $\det(Z_i) \geq \det(Z_{i-1})$. We write

$$\sum_{i=1}^t f_i^\top Z_i^{-1} f_i = \sum_{i=1}^t 1 - \frac{\det(Z_{i-1})}{\det(Z_i)} \leq \sum_{i=1}^t \log\left(\frac{\det(Z_i)}{\det(Z_{i-1})}\right) = \log\left(\frac{\det(Z_t)}{\det(Z_0)}\right),$$

where we used the fact that $1 - x \leq \log(1/x)$ for $x \leq 1$. ∎

We are now ready to prove Lemma 1.

*Proof of Lemma 1.* The first claim is already proved in Lemma F.4. We focus on proving the second claim. Recall the result of Lemma F.9, which states that

$$\sum_{t=1}^T f_t^\top Z_t^{-1} f_t \leq \log\left(\frac{\det(Z_T)}{\det(\alpha I)}\right).$$

Using matrix determinant lemma, the above is equivalent to

$$\sum_{t=1}^T \frac{f_t^\top Z_{t-1}^{-1} f_t}{1 + f_t^\top Z_{t-1}^{-1} f_t} \leq \log\left(\frac{\det(Z_T)}{\det(\alpha I)}\right).$$

By Lemma F.1, $\log \det(Z_t)$ is bounded by $\text{polylog}(T)$ with high probability since $k \asymp_M \text{polylog}(T)$. Furthermore, by Lemma F.8, $\|Z_{t-1}^{-1/2} f_t\|_2^2 \lesssim_M \text{polylog}(T)$ with high probability. Concretely,

$$\mathbb{P}(\|Z_{t-1}^{-1/2} f_t\|_2^2 \leq 10\kappa(2mk + 4\log(2/\delta))/p^2) \geq 1 - \delta,$$

$$\mathbb{P}\left(\log(\det(Z_t)) \leq mk \log\left[\alpha^2 + 8k(R_P^2 + 1)(R_x^2 + R_C)(1 + \gamma)^4(mt - \log(\delta))t^{3+2\log(\gamma)}\right]\right) \geq 1 - \delta.$$

Therefore, we can apply Lemma G.2 by combining the two bounds and taking a union bound

$$R_Z(T) \triangleq mk \log \Big[ \alpha^2 + 8k(R_P^2 + 1)(R_x^2 + R_C)(1+\gamma)^4(mT - \log(\delta))T^{3+2\log(\gamma)} \Big],$$

$$\mathbb{P}\Big\{ \sum_{t=1}^{T} \|Z_{t-1}^{-1/2}f_t\|_2^2 \le \Big(1 + \frac{10\kappa(2mk + 4\log(4/\delta))}{p^2}\Big)\big(R_Z(T) - mk\log(\alpha)\big) \Big\} \ge 1 - \delta.$$

$\square$

## F.5 Regularization term

The following lemma computes an upper bound on the 2-norm of the relaxed model parameters $\widetilde{\Theta}$.

**Lemma F.10. (Model parameter bound)** *Consider system (4) and let $k$ be the number of spectral filters and $\widetilde{\Theta}$ be the parameters defined in (22). If $\|\mathcal{O}_t\|_2, \|\mathcal{C}_t\|_2 \le R_K$ and $\|D\|_2 \le R_P$ then,*

$$\|\widetilde{\Theta}\|_2 \le 2kR_K + R_P.$$

*Proof.* Parameter matrix $\widetilde{\Theta}$ is the concatenation of coefficients of features $\widetilde{y}_{t-1}, \widetilde{x}_{t-1}, x_t$. By matrix norm properties,

$$\|\widetilde{\Theta}\|_2 \le \|D\|_2 + \sum_{j=1}^{k} \|\sum_{i=1}^{d} Cv_i w_i^\top K\langle\mu(\lambda_i), \phi_j\rangle\|_2 + \|\sum_{i=1}^{d} Cv_i w_i^\top (B - KD)\langle\mu(\lambda_i), \phi_j\rangle\|_2.$$

Recall that $\{\lambda_i\}_{i=1}^{k}$, $\{v_i\}_{i=1}^{k}$, and $\{w_i^\top\}_{i=1}^{k}$ are the top $k$ eigenvalues, right eigenvectors, and left eigenvectors of $G$, respectively. Write

$$\|\sum_{i=1}^{d} Cv_i w_i^\top K\langle\mu(\lambda_i), \phi_j\rangle\|_2 = \|\sum_{t=1}^{T} CG^{T-t}K\phi_j(t)\|_2 = \|\mathcal{O}_T\phi_j\|_2 \le R_K,$$

and similarly,

$$\|\sum_{i=1}^{d} Cv_i w_i^\top (B - KD)\langle\mu(\lambda_i), \phi_j\rangle\|_2 = \|\mathcal{C}_T\phi_j\|_2 \le R_K.$$

Summing all terms gives the final bound. ∎

**Lemma F.11. (Regularization term bound)** *Assume as in Lemma F.10 and let $Z_t = \alpha I + f_t f_t^\top$. If $\alpha \le 1/\|\widetilde{\Theta}\|_2^2$, then*

$$\|\alpha\widetilde{\Theta}Z_{t-1}^{-1/2}\|_2^2 \le 1.$$

*Proof.* The regularization term implies $Z_t \succeq \alpha I$ and thus $\|Z_t^{-1/2}\|_2^2 \le 1/\alpha$. By norm properties

$$\|\alpha\widetilde{\Theta}Z_{t-1}^{-1/2}\|_2^2 \le \alpha^2\|\widetilde{\Theta}\|_2^2\|Z_{t-1}^{-1/2}\|_2^2 \le 1.$$

∎

## F.6 Innovation error

The following lemma, based on the analysis given by Tsiamis and Pappas [11], shows that the innovation error is bounded by $\sqrt{\mathcal{L}(T)}$ (defined in (25)).

**Lemma F.12. (Innovation error bound)** *Let $\mathcal{L}(T) = \sum_{t=1}^{T} \|\hat{m}_t - m_t\|_2^2$ be the squared error between Kalman predictions in hindsight and predictions by Algorithm 1. Assume that the innovation covariance matrix has a bounded norm $\|V\|_2 \le R_V$. For all $\delta > 0$, the following holds with probability greater than $1 - \delta$:*

$$\sum_{t=1}^{T} 2e_t^\top(\hat{m}_t - m_t) \le 8R_V^2\Big(\mathcal{L}(T) + 1\Big)^{1/2}\Big[2 + \log\Big(\frac{\mathcal{L}(T) + 1}{\delta}\Big)\Big].$$

*Proof.* Write

$$\sum_{t=1}^{T} e_t^\top (\hat{m}_t - m_t) = \sum_{t=1}^{T} \sum_{i=1}^{m} e_{t,i}(\hat{m}_{t,i} - m_{t,i}).$$

Let $s = m\lfloor s/m \rfloor + r$ and define the following filtration

$$\mathcal{F}_s = \{e_{1,1}, \ldots, e_{\lfloor s/m \rfloor, r}\}.$$

A scalar version of Theorem F.1 states that the following holds with probability at least $1 - \delta$

$$\Big(\sum_{t=1}^{T} \|\hat{m}_t - m_t\|_2^2 + 1\Big)^{-1/2} \sum_{t=1}^{T} e_t^\top (\hat{m}_t - m_t) \le 4R_V^2 \Big[2 + \log\Big(\frac{1}{\delta}\Big) + \log\Big(\sum_{t=1}^{T} \|\hat{m}_t - m_t\|_2^2 + 1\Big)\Big].$$

Therefore, with probability at least $1 - \delta$

$$\sum_{t=1}^{T} 2e_t^\top (\hat{m}_t - m_t) \le 8R_V^2 \Big(\mathcal{L}(T) + 1\Big)^{1/2} \Big[2 + \log\Big(\frac{\mathcal{L}(T) + 1}{\delta}\Big)\Big].$$

∎

## F.7 Proof of Theorem 1

*Proof of Theorem 1.* Recall the regret decomposition given in Appendix E:

$$\text{Regret}(T) \le \sup_{1 \le t \le T} \Big( \|E_{t-1} Z_{t-1}^{-1/2}\|_2^2 + \|B_{t-1} Z_{t-1}^{-1/2}\|_2^2 + \|\alpha \widetilde{\Theta} Z_{t-1}^{-1/2}\|_2^2 \Big) \Big( \sum_{t=1}^{T} \|Z_{t-1}^{-1/2} f_t\|_2^2 \Big)$$

$$+ T \sup_{1 \le t \le T} \|b_t\|_2^2 - \sum_{t=1}^{T} 2e_t^\top (\hat{m}_t - m_t).$$

Let $\delta_1 = \delta/8$. We describe bounds on each term in the above regret bound. All lemmas and theorems used in this proof contain explicit dependencies on horizon $T$ as well as PAC bound parameters. While one can combine these results to write a regret bound with explicit dependencies on all parameters, we refrain from writing in such detail here for a clear presentation.

**Bounding $\|\mathbf{E_{t-1} Z_{t-1}^{-1/2}}\|_2^2$.** According to Theorem F.1, with probability at least $1 - \delta_1$, the term $\|E_{t-1} Z_{t-1}^{-1/2}\|_2^2$ is bounded by

$$\|E_{t-1} Z_{t-1}^{-1/2}\|_2 \lesssim \text{poly}(R_\Theta, m) \Big[ \log(1/\delta_1) + \log(\det(Z_t)) - l \log(\alpha) \Big],$$

$l = (m + n)k + n$ is the feature vector dimension. We substitute the regularization parameter $\alpha$ and the number of filters $k$ according to Theorem 1 assumption (iii). Given the values for $k, \alpha$ and by Lemma F.1, with probability at least $1 - \delta_1$ we have

$$\log(\det(Z_t)) \lesssim \text{poly}(R_\Theta, m, \beta) \, \text{polylog}(\gamma, \frac{1}{\delta_1}) \log^3(T).$$

Taking a union bound gives

$$\mathbb{P}\Big[\|E_{t-1} Z_{t-1}^{-1/2}\|_2^2 \lesssim \text{poly}(R_\Theta, m, \beta) \, \text{polylog}(\gamma, \frac{1}{\delta_1}) \log^6(T)\Big] \ge 1 - 2\delta_1. \tag{30}$$

**Bounding $\|\mathbf{B_{t-1} Z_{t-1}^{-1/2}}\|_2^2$.** Recall the definitions $B_t = \sum_{i=1}^{t} b_i f_i^\top$ from (26) and $b_i = \widetilde{\Theta} f_t - m_t$ from (24). We choose the number of filters $k$ to satisfy (23) with failure probability $\delta_1 > 0$ and $\epsilon = 1/T$,[1] which results in $k \gtrsim_M \log^2(T)$ satisfied by assumption (iii). Therefore, we can apply

Theorem 3 which states that $\|b_t\|_2^2 \leq 1/T$ with probability at least $1 - \delta_1$. Combining this result with the result of Theorem F.1 with a union bound yields

$$\mathbb{P}\Big[\|\Big(\sum_{i=1}^{t-1} b_i f_i^\top\Big) Z_{t-1}^{-1/2}\|_2 \leq \frac{4}{T}\big(2m + \log\big(\frac{\det(Z_t)^{1/2}\det(\alpha I_l)^{-1/2}}{\delta_1}\big)\big)\Big] \geq 1 - 2\delta_1.$$

With a similar argument used in bounding $\|E_{t-1} Z_{t-1}^{-1/2}\|_2^2$, we have

$$\mathbb{P}\Big[\|B_{t-1} Z_{t-1}^{-1/2}\|_2^2 \lesssim \operatorname{poly}(R_\Theta, m, \beta)\operatorname{polylog}(\gamma, \frac{1}{\delta_1})\frac{\log^6(T)}{T}\Big] \geq 1 - 3\delta_1. \qquad (31)$$

**Bounding $\|\alpha\widetilde{\Theta} \mathbf{Z_{t-1}^{-1/2}}\|_2^2$.** By assumption (iii) and as a result of Lemma F.11, we have

$$\|\alpha\widetilde{\Theta} Z_{t-1}^{-1/2}\|_2^2 \lesssim 1.$$

**Bounding $\sum_{\mathbf{t=1}}^{\mathbf{T}} \|\mathbf{Z_{t-1}^{-1/2}} \mathbf{f_t}\|_2^2$.** Lemma 1 provides the following bound on the excitation term

$$\mathbb{P}\Big[\sum_{t=1}^{T} \|Z_{t-1}^{-1/2} f_t\|_2^2 \lesssim \kappa\operatorname{poly}(R_\Theta, m, \beta)\operatorname{polylog}(\gamma, \frac{1}{\delta_1})\log^5(T)\Big] \geq 1 - \delta_1, \qquad (32)$$

where the number filters $k$ is substituted by assumption (iii).

**Bounding $\mathbf{T}\sup_{\mathbf{1\leq t\leq T}} \|\mathbf{b_t}\|_2^2$.** Applying Theorem 3 with parameters $\delta_1 > 0, \epsilon = 1/T$, we have

$$\mathbb{P}\Big[T\sup_{1\leq t\leq T} \|b_t\|_2^2 \leq T\epsilon \leq 1\Big] \geq 1 - \delta_1. \qquad (33)$$

Recall from Appendix E that $\mathcal{L}(T)$ is bounded by

$$\mathcal{L}(T) \leq \sup_{1\leq t\leq T}\Big(\|E_{t-1} Z_{t-1}^{-1/2}\|_2^2 + \|B_{t-1} Z_{t-1}^{-1/2}\|_2^2 + \|\alpha\widetilde{\Theta} Z_{t-1}^{-1/2}\|_2^2\Big)\Big(\sum_{t=1}^{T} \|Z_{t-1}^{-1/2} f_t\|_2^2\Big) + T\sup_{1\leq t\leq T} \|b_t\|_2^2.$$

Lemma F.12 with $\delta_1$ states that

$$\mathbb{P}\Big[\sum_{t=1}^{T} e_t^\top(\hat{m}_t - m_t) \lesssim \operatorname{poly}(R_\Theta)\operatorname{polylog}\Big(\frac{1}{\delta_1}\Big)\sqrt{\mathcal{L}(T) + 1}\Big] \geq 1 - \delta_1. \qquad (34)$$

Combining the bounds given in (30), (31), (32), (33), (34), taking a union probability bound, and setting $\delta = 8\delta_1$ gives

$$\mathbb{P}\Big[\operatorname{Regret}(T) \leq \kappa\log^{11}(T)\operatorname{poly}(R_\Theta, \beta, m)\operatorname{polylog}(\gamma, \frac{1}{\delta})\Big] \geq 1 - \delta.$$

$\square$

# G  Auxiliary lemmas

In this section, we present a few lemmas that we use throughout the theoretical analysis of our algorithm, presented here for completeness.

The following lemma provides an upper bound on the norm of block Toeplitz matrices [12].
**Lemma G.1. (Triangular Block Toeplitz Norm)** *Let $\mathcal{T}_i \in \mathbb{R}^{m_1, m_2}$ for $i = 1, 2, \ldots, n$. Define the following triangular block Toeplitz matrix*

$$\mathcal{T} = \begin{bmatrix} \mathcal{T}_1 & \mathcal{T}_2 & \mathcal{T}_3 & \ldots & \mathcal{T}_{n-1} & \mathcal{T}_n \\ 0 & \mathcal{T}_1 & \mathcal{T}_2 & \ldots & \mathcal{T}_{n-2} & \mathcal{T}_{n-1} \\ \vdots & & & & & \\ 0 & 0 & 0 & \ldots & \mathcal{T}_1 & \mathcal{T}_2 \\ 0 & 0 & 0 & \ldots & 0 & \mathcal{T}_1 \end{bmatrix}.$$

*Then,*

$$\|\mathcal{T}\|_2 \leq \sum_{i=1}^{n} \|\mathcal{T}_i\|_2.$$

The following is a simple result for upper bounding a series.

**Lemma** G.2. *Let $t \in \mathbb{N}$ and let $z_t$ to be a non-negative sequence bounded by a non-decreasing poly-logarithmic function $g(t)$. Suppose that the following sum*

$$\sum_{t=1}^{T} \frac{z_t}{1 + z_t}$$

*is bounded by $h(T)$, a non-decreasing poly-logarithmic function of $T$. Then, $\sum_{t=1}^{T} z_t$ is bounded by a non-decreasing function poly-logarithmic in $T$.*

*Proof.* Let $z_m = \max_{t \in \{1,\dots,T\}} z_i$. We have $z_m \leq g(m) \leq g(T)$. Therefore,

$$\sum_{t=1}^{T} z_t \leq \sum_{t=1}^{T} \frac{1 + z_m}{1 + z_t} z_t \leq (1 + g(T)) \sum_{t=1}^{T} \frac{z_t}{1 + z_t} \leq (1 + g(T)) h(T), \tag{35}$$

which is the desired conclusion. ∎

# H    Additional experiments

**Comparison with the EM algorithm.**    We conduct an experiment in a scalar LDS to compare the performance of our algorithm with the EM algorithm that estimates system parameters (Figure 1, left). The parameters estimated by the EM algorithm are later used by the Kalman filter for predictions. In this experiment, we set the horizon $T = 200$ due to the large computation time required by the EM algorithm. The number of filters $k$ is set to 5 for all other three algorithms. The experiment was simulated 100 independent times and the average error together with the 99% confidence intervals are presented.

Figure 1: Left: Performance of our algorithm compared with wave filtering, truncated filtering, and expectation maximization in a scalar system with parameters $A = B = C = D = 1$, noise covariance matrices $Q = R = 0.001$, inputs $x_t \sim \mathcal{N}(0, 2)$, and horizon $T = 200$. Right: Hyperparameter sensitivity of our algorithm in the same systems with inputs $x_t \sim \mathcal{N}(0, 0.5)$ and horizon $T = 10000$.

For the system considered in this experiment, EM performs poorly. System-identification-based methods such as EM, besides being significantly slower, do not have regret guarantees and they can fail in some examples; a similar observation was made by Hazan et al. [4].

**On hyperparameters.**    The SLIP algorithm has two hyperparameters: the number of filters $k$ and the regularization parameter $\alpha$. In the experiments, we set $\alpha > 0$ only when the empirical feature covariance matrix is singular, which we observe only happens in the first two time steps. For the number of filters $k$, Theorem 1 provides a guideline of choosing $k$ of order $\log^2(T)$. The right plot in Figure 1 demonstrates the sensitivity of the SLIP algorithm with respect to the number of filters $k$. The system considered for this experiment is scalar with Gaussian inputs and the horizon is set to 10000. As before, the experiment was simulated 100 independent times. We vary $k$ from 5 to 35 and measure the average prediction error from 5000 to 10000 ($N = 5000$ in the plot). We observe that the SLIP algorithm is robust with respect to parameter $k$.

# I  Systems with long forecast memory

As discussed in the paper, system (4) exhibits long forecast memory when $\rho(G)$ is close to one. The closed-loop matrix $G$ itself is related to parameters $A, C, Q$, and $R$. In the following example, we discuss when long forecast memory is instantiated in a scalar dynamical system.

**Example** I.1. *Consider system (4) with $d = m = 1$. The following holds for a stationary Kalman filter*

$$KC = \frac{AC^2P^+}{C^2P^+ + R} \Rightarrow 0 \leq KC \leq A \qquad for \ d = m = 1,$$

*where $P^+$ is the variance of state predictions $\hat{h}_{t|t-1}$ [6]. The above constraint yields $G = A - KC \leq A$, which implies that the forecast memory can only be long in systems that mix slowly. We write*

$$G = A\big(1 - \frac{C^2P^+}{C^2P^+ + R}\big), \qquad for \ d = m = 1.$$

*The above equation suggests if $R \gg C^2P^+$, then $G$ is close to $A$. In words, linear dynamical systems with small observed signal to noise ratio $C/\sqrt{R}$ have long forecast memory, provided that they mix slowly.*

*Another parameter that affects the forecast memory of a system is the process noise variance $Q$. When $Q$ is small and $A$ is close to one, latent state $h_t$ is almost constant. In this setting, the observations in the distant past are informative on $h_t$ and therefore should be considered when making predictions.*

In multi-dimensional systems, the chance of encountering a system with long forecast memory is much higher as it suffices for only one variable or direction to exhibit long forecast memory. Systems represented in the discrete-time form of Equation (4) are often obtained by discretizing differential equations and continuous dynamical systems, for which choosing a small time step results in a better approximation. However, reducing the time step directly increases the forecast memory. These types of issues has motivated a large body of research on alternative methods such as continuous models [9] and adaptive time steps [2]. It is therefore desirable to have algorithms whose performance is not affected by the choice of time step, which is one of our goals in this paper.

## Footnotes

[1]Setting $\epsilon = 1/T$ is later used for a uniform bound on $\|b_t\|_2^2$ and is not critical in this part of the proof.