[Reviews · NeurIPS 2020]

Review 1

Summary and Contributions: Authors derive novel online convex optimisation approaches to improper learning of LDS, which provide logarithmic regret guarantees even in the case of a long forecast memory.

Strengths: System identification has recently seen something of a Renaissance, with several major recent results appearing, incl. those of Hazan et al (NIPS 2017, 2018), Kozdoba et al (AAAI 2019), and Tsiamis and Pappas (submitted). This is a fundamental contribution, combining many non-trivial ideas including the spectral filtering of Hazan et al and truncations therein of Kozdoba et al and Tsiamis and Pappas, and substantially improving upon all the recent works, which made many simplifying assumptions. A key realisation is the "regret decomposition", followed by many technical steps to bound each part in the decomposition. The regret decomposition could be very useful, more generally. The results rely on Kolmogorov widths, a rather non-trivial branch of approximation theory. These can be used much more widely in statistical theory and machine learning.

Weaknesses: The experimental part is very weak, for a number of reasons: -- the examples chosen do not meet the requirements of the other methods tested. (Notably, the non-symmetric matrix A of System 3.) So the authors are beating the straw man, in some sense. -- the source code of the authors has not been attached. It's not possible to verify the results and recreate the plots. -- While I appreciate that the authors wanted to pick examples with long forecast memory, their choice seems rather arbitrary. (Notably System 2 has a lot of arbitrary looking constants.) It would be good to present some empirical perturbation analysis, showing that the results of the proposed method are "no fluke". -- in the case of the work of Hazan et al, it's not clear whether the users utilise the Wave filter or the Spectral filter, actually, or how do they implement it, considering there is no official implementation released. Both methods are non-trivial to implement. The reliance on Kolmogorov widths, e.g. the results of Temlyakov, https://link.springer.com/content/pdf/10.1007/BF02312773.pdf is not made very clear. More generally, even Theorem 2 in the main body of the text may be borderline incomprehensible without a background in approximation theory that cannot be expected in the NeurIPS community at large. ("Five people in the world"?)

Correctness: As far as I can tell.

Clarity: While the first five pages are very clear, Theorem 2 in the main body of the text may be borderline incomprehensible without a background in approximation theory that cannot be expected in the NeurIPS community at large. ("Five people in the world"?)

Relation to Prior Work: Several major recent results are discussed in detail, with a considerable insight, incl. those of Hazan et al (NIPS 2017, 2018), Kozdoba et al (AAAI 2019), and Tsiamis and Pappas (submitted). While the present paper combines non-trivial ideas including the spectral filtering of Hazan et al and truncations therein of Kozdoba et al and Tsiamis and Pappas, it credits them fairly and substantially improves upon all the recent works, which made many simplifying assumptions. The paper does not mention recent works by Maryam Fazel: https://scholar.google.com/citations?hl=en&user=vlN_kRoAAAAJ&view_op=list_works&sortby=pubdate and Ben Recht: https://scholar.google.com/citations?hl=en&user=a_dbdxAAAAAJ&view_op=list_works&sortby=pubdate and Csaba Szepesvari: https://scholar.google.com/citations?hl=en&user=zvC19mQAAAAJ&view_op=list_works&sortby=pubdate While it does not seem to utilise any particular ideas from these, the citations may be welcome.

Reproducibility: No

Additional Feedback: I have read the rebuttal and I am happy with the answers.


Review 2

Summary and Contributions: The paper "Spectral Kalman Filtering: Learning to Predict in Unknown Dynamical Systems with Long-Term Memory" provides an algorithm to approximate Kalman filter learning for online prediction of unknown and partially observed LDS. The main contributions are four-fold, they show: 1) that for diagonizable forecast memory G with real eigenvalues the Kalman filter coefficients can be approximated by a linear combination of known filters, 2) the general difficulty of approximating Kalman filter coefficients via linear subspaces in general, 3) a logarithmic regret bound for the proposed algorithm when the system is marginally stable, and 4) experimental results in 3 simulations of LDS prediction.

Strengths: The contribution is well-structured, well-written and theoretically founded. The theorems are well explained and set into context with respect ot the state-of-the-art. As fast approximation algorithm this can be a useful tool for practical use in 1 step forecasting with provable risk bounds for the class of LDS satisfying the assumptions. The exhaustive rebuttal was one of the best I have ever read. The authors addressed all reviewer doubts and suggestions very convincingly and I changed my assessment acknowledging that this contribution is among the top in NeurIPS.

Weaknesses: It would have been nice to give real world examples in the experiments for systems where the assumptions for the approximation bounds hold to demonstrate the usefulness in real-world scenarios. Also the influence of the choice of the hyperparameters and their sensitivities could have been shown.

Correctness: This paper is mainly theoretical. The theorems seem to be correct. The empirical contribution is not the main focus.

Clarity: Yes it is.

Relation to Prior Work: It differs in its analysis based on the forecast memory and approximation by spectral methods

Reproducibility: No

Additional Feedback: Theoretically yes, empirically one would need transparent list of hyperparameter settings used for all methods, but the practical side was not the focus here. It would have been nice though, since the contribution claims increased practical use in comparison to existing techniques, so more detailed demonstration also on real-world scenarios would be much more convincing than just synthetic examples.


Review 3

Summary and Contributions: This paper proposes an efficient online prediction algorithm for unknown linear dynamical systems with the long-term prediction memory property. Motivations and contributions are very clearly written.

Strengths: Derivation of the proposed algorithm using some techniques such as Kolmovorov width and small-ball condition is theoretically sound.

Weaknesses: Experiment is a little weak. It would be more helpful, if you showed more exhaustive comarison results. Personally, I want to know the comparison with the usual Kalman filtering with identified model parameters.

Correctness: Yes, the claims made by the authors seem correct.

Clarity: Yes, this paper is well written.

Relation to Prior Work: Basically, yes. But, I am not clear whether the proposed algorithm (Algorithm 1) is identical to the spectral filtering by [Hazan 2017] or not. What is the main difference between them ? The authors clearly answered to my question. Now I understood the difference between them.

Reproducibility: Yes

Additional Feedback: While the authors named the proposed method "spectral Kalman filtering", I feel it is a little misleading because Kaman filtering algorithm means recursive update of the posterior distribution (mean vector and covariance matrix) of the latent state vector for many people. Although it is no doubt the proposed method has a close connection to Kalman filtering, it doesn't explicitly estimate the posterior of state vector.


Review 4

Summary and Contributions: The paper presents a model-free algorithm that converges to Kalman Filter (optimal prediction) results in polynomial time. The predictor is shown to work for marginally stable systems and avoids the inferior performance of a fixed-length window to estimate the future observations. The paper provides a statistical as well as geometric reasoning to justify the technical results. The performance is also verified with an experimental setting where the methodology is compared with two other recently proposed algorithms.

Strengths: The paper shows a strong background on control theory and statistical learning. The topic of the paper is timely and ideas are original. The main result of the paper sounds authentic and with adequate theoretical grounding. The empirical plots also give a good sense of the performance of the algorithm. The contribution is significant and of high relevance to the statistical learning and controls communities.

Weaknesses: The paper begins with a general scope of the problem setup and boils down to the control-free settings for the main results. While this assumption is reasonable due to the space limitation and the amount of analysis behind the general setup, the manuscript would be more complete in case of having a note on the difficulties of adding (active) control. For instance, removing the control from the setup makes the excitation assumption almost trivial (based on martingale small ball condition), but this would definitely need more investigation when the control design problem is added to the scenario.

Correctness: The claims, results, and empirical methodology sound correct.

Clarity: Overall, the paper is well written and easy to follow. There are only a few instances that the authors might want to reconsider. For example B_t is used in the analysis while B was initially reserved for the input. Adding dimensions when introducing new variables (such as Psi in the algorithm) also makes the paper more readable.

Relation to Prior Work: It is clearly explained how the paper is different from previous works in the literature. The authors are also suggested to cite a textbook or a seminal paper on the original Kalman Filter theory.

Reproducibility: Yes

Additional Feedback: After going over the rebuttal, I believe the authors are already aware of the shortcomings of the paper and they intend to add discussions to the manuscript to show that. Regarding my point on excitation signals, as the authors pointed out, having control in the setup makes the machinery more complicated and I agree. I believe the work has the potential to be extended to the case with input analysis. In general, I firmly believe that the paper has a very well-organized structure and the proofs are rigorously provided and I stick to my score.

[Author Response · NeurIPS 2020]

We thank the reviewers for their thorough review and insightful feedback and appreciate their positive comments.
**Experiments:** We agree with the reviewers that further empirical evaluation of our algorithm spectral Kalman filtering
(SKF) is beneficial. We will add more experiments to the final manuscript. Here we present our preliminary work on
these experiments. Minor details on experiments are excluded due to space constraint.
(1) *Hyperparameter sensitivity:* The hyperparameters of SKF are the number of filters $k$ and the regularization parameter
$\alpha$. In the paper, we set $k = 20$ and use $\alpha > 0$ (in particular, $\alpha = 10^{-4}$) when the empirical feature covariance is singular,
which we observe only happens in the first two time steps. Per Reviewer 2's suggestion, we analyzed the hyperparameter
sensitivity of our algorithm. Figure 1(a) shows that our algorithm is robust with respect to hyperparameter $k$.
(2) *Perturbation analysis:* Per Reviewer 1's suggestion, we analyze the robustness of SKF in the presence of nonlinear
perturbation according to logistic growth dynamics $h_{t+1} = Ah_t - \beta h_t^2 + Bx_t + \eta_t$. We vary $\beta$ and choose $x_t$ such
that observations do not explode. Figure 1(b) illustrates average prediction error vs. $\beta$ that shows the performance of
our algorithm degrades gracefully with non-linear perturbation.
(3) *Comparison with system identification (SI) methods:* Per Reviewer 3's suggestion, we plan to compare our algorithm
with SI-based methods for the final paper. Here we present one experiment comparing the performance with EM
followed by Kalman filtering (Figure 1(c)). In addition, we showed superior performance over wave filtering of Hazan
et al. (2017), who demonstrated that their algorithm works better than SI-based methods. SI-based methods, besides
being often significantly slower, either do not have regret guarantees or have degrading performance when $\rho(A) \to 1$.
**Reviewer 1:** [*Non-symmetric $A$*] We compare the algorithms in both systems with symmetric and non-symmetric $A$.
Although the theoretical analysis of wave filtering is given for symmetric $A$, the authors mention in their paper that
empirically their algorithm also handles non-symmetric $A$. System 3 experiment compares the performance of wave
filtering when $A$ is non-symmetric and $\rho(A) = 1$, which results in a growing observation norm. [*System parameters*]
For System 2, we set the parameters somewhat arbitrary to encounter a system that exhibits long memory. While
the spectral norm of $A$ and $G$ are parameters of interest that distinguish the performance of algorithms, we were
not concerned with particular values of other parameters. [*Perturbation analysis*] We thank the reviewer for their
suggestion. We have included an experiment here and will include further analysis to the updated manuscript. [*Wave
filter implementation*] While we were not provided with the source code, Hazan et al. (2017) gives a detailed explanation
of their algorithm in their paper and appendix. We implemented wave filter with follow the leader algorithm (as opposed
to projected gradient descent) as suggested in their paper for better performance and by our observations that tuning the
step size for gradient descent implementation of wave filter was very difficult. [*Source code*] We plan to make the code
public after optimizing the implementations. [*Clarity*] We agree with the reviewer that the clarity of the approximation
theory section can be improved. We will draw connections with Temlyakov's results and elaborate more when updating
the manuscript to make it as accessible as possible. [*Citations*] We thank the reviewer for recommending additional
references. We will include them in the final paper.
**Reviewer 2:** [*Real-world experiments*] We agree with the reviewer that testing our algorithm on real applications is
important. We are collaborating with a team to apply the proposed algorithm to healthcare where long-term forecast
memory is critical. [*Hyperparameters*] We have provided the hyperparameter $k$ in the paper and have discussed $\alpha$
above. We agree that analyzing hyperparameter sensitivity is critical for practical considerations and will be included.
**Reviewer 3:** [*Experiments*] We thank the reviewer for their suggestion and refer to the above segment on experiments.
[*Comparison with Hazan et al. (2017)*] Our algorithm shares some similarities with wave filter but has important
differences: (i) We design our filters by applying spectral methods to Kalman predictions which results in different
filters compared to those in wave filter (ours is related to the covariance of $[1, \lambda, \dots, \lambda^T]$ for $\lambda \in [-1, 1]$ while theirs
is related to $[1 - \lambda, \dots, \lambda^{T-1} - \lambda^T]$ for $\lambda \in [0, 1]$). This filter construction poses many theoretical challenges such
as growing feature norm and Lipschitz constant but is crucial for convergence to Kalman prediction which was left
open in Hazan et al. (2017). (ii) Wave filter only includes the most recent observation but our algorithm considers all
past observations. (iii) Wave filter is based on projected gradient descent whereas our algorithm uses regularized least
squares. [*Algorithm name*] We concur with the reviewer and consider a more descriptive name (such as Spectral LDS
Adaptive Predictor). We are also working on a recursive version and hope to achieve that for the final version.
**Reviewer 5:** [*Control inputs*] While the algorithm derivation, improper learning approximation error, and most of the
regret analysis consider control inputs, the excitation result is given without inputs. We believe that extending our
analysis for LDS with inputs is possible by characterizing input features and in light of the experiments. As pointed out
by the reviewer, such an extension requires some care. For instance, one needs to characterize the covariance between
features constructed from observations and features constructed from inputs. We will add a discussion to the final paper.
[*Notation and citation*] We thank the reviewer for their suggestions and will incorporate them in the final paper.

Figure 1: Experiments on hyperparameter sensitivity (a), pertubation analysis (b), comparison with EM (c).

[Meta-Review · NeurIPS 2020]

This is a well-written paper addressing an important and timely problem. The authors rebuttal was well done, and managed to answer lingering doubts of the reviewers. The consensus among the reviewers after the rebuttal was that the paper be accepted. Adding another point to the reviewers' comments: One remaining issue in my opinion, and a limitation of the results, is that the assumption in theorem 1 that G has real eigenvalues is far more restrictive than it seems: a system with only teal eigenvalues cannot capture many interesting physical dynamics; for example any system whose response has any kind of oscillation needs complex eigenvalues. So extension to nonsymmetric G is needed and I strongly encouraged the authors to pursue it for the work to be more useful; the later work of Hazan manages to do this for the wave filter approach. Still, the results have enough novelty to be published and will contribute to NeurIPS community.